# The recycling endosome protein Rab25 coordinates collective cell movements in the zebrafish surface epithelium

Patrick Morley Willoughby[1], Molly Allen[1], Jessica Yu[2,3†], Roman Korytnikov[1], Tianhui Chen[1], Yupeng Liu[1], Isis So[1], Haoyu Wan[1], Neil Macpherson[1], Jennifer A Mitchell[1], Rodrigo Fernandez-Gonzalez[1,2,3], Ashley EE Bruce[1]*

[1]Department of Cell and Systems Biology, University of Toronto, Toronto, Canada; [2]Ted Rogers Centre for Heart Research, Translational Biology and Engineering Program, University of Toronto, Toronto, Canada; [3]Institute of Biomaterials and Biomedical Engineering, University of Toronto, Toronto, Canada

**Abstract** In emerging epithelial tissues, cells undergo dramatic rearrangements to promote tissue shape changes. Dividing cells remain interconnected via transient cytokinetic bridges. Bridges are cleaved during abscission and currently, the consequences of disrupting abscission in developing epithelia are not well understood. We show that the Rab GTPase Rab25 localizes near cytokinetic midbodies and likely coordinates abscission through endomembrane trafficking in the epithelium of the zebrafish gastrula during epiboly. In maternal-zygotic Rab25a and Rab25b mutant embryos, morphogenic activity tears open persistent apical cytokinetic bridges that failed to undergo timely abscission. Cytokinesis defects result in anisotropic cell morphologies that are associated with a reduction of contractile actomyosin networks. This slows cell rearrangements and alters the viscoelastic responses of the tissue, all of which likely contribute to delayed epiboly. We present a model in which Rab25 trafficking coordinates cytokinetic bridge abscission and cortical actin density, impacting local cell shape changes and tissue-scale forces.

*For correspondence:
ashley.bruce@utoronto.ca

Present address: †School of Biomedical Engineering, University of British Columbia, Vancouver, Canada

Competing interests: The authors declare that no competing interests exist.

## Introduction

During metazoan development, epithelial cells are often organized into cohesive sheets that undergo large-scale cellular movements. In proliferative tissues, cell division is fundamental to the establishment of normal tissue architecture (*Gibson et al., 2006*). Cell division also poses a challenge for developing epithelia, as dividing cells must maintain cell-cell contacts while simultaneously coordinating shape changes with surrounding cells (*Higashi et al., 2016*). In the zebrafish embryo, the single-cell thick surface epithelium, the enveloping layer (EVL), undergoes rounds of proliferation during epiboly, a gastrulation movement describing the expansion and spreading of a tissue (*Campinho et al., 2013*). Thus, the zebrafish gastrula is an ideal system to investigate epithelial morphogenesis during gastrulation.

The multilayered zebrafish blastoderm is positioned on top of a large yolk cell. During epiboly, the blastoderm spreads vegetally to completely cover the yolk cell (reviewed in *Bruce and Heisenberg, 2020*). Epiboly initiates at dome stage when EVL spreading is triggered by a reduction in tissue surface tension, that enables vegetal directed tissue expansion (*Morita et al., 2017*). EVL cells thin apicobasally as the apical surface area expands, with cell divisions preferentially oriented along the animal-vegetal axis (*Campinho et al., 2013*). EVL proliferation slows as the blastoderm covers half of the yolk cell at 50% epiboly (*Campinho et al., 2013*).

At 6hpf, a contractile actomyosin ring assembles in the region of the yolk cell adjacent to the blastoderm, the yolk syncytial layer (YSL) (*Behrndt et al., 2012*). Contractile forces generated by the

ring tow the tightly attached EVL vegetally. During the later stages of epiboly, as EVL cells continue to elongate and spread, EVL tension is highest, particularly in marginal regions closest to the yolk cell (*Campinho et al., 2013*). A subset of marginal EVL cells actively change shape and rearrange to leave the margin, enabling the tissue circumference to narrow and eventually close at the vegetal pole (*Köppen et al., 2006*). Overall, EVL morphogenesis involves coordination between local cell shape changes, rearrangements and tissue-scale forces.

Cell divisions play a critical and conserved role during epithelial morphogenesis in multiple systems (*Campinho et al., 2013*; *Firmino et al., 2016*; *Lau et al., 2015*; *Wyngaarden et al., 2010*). Following division, daughter cells either share a common interface or cell rearrangements occur (*Firmino et al., 2016*; *Gibson et al., 2006*). Both events can be viewed as multicellular processes as cell division impacts both the dividing cell and adjacent neighbors. For example, during the formation of de novo cell contacts between newly formed daughter cells, neighboring cells resist contractile forces to maintain shape as shown in the *Xenopus* gastrula and *Drosophila* pupal notum (*Herszterg et al., 2013*; *Higashi et al., 2016*).

Cell divisions also impact tissue scale forces. Oriented cell divisions have well-characterized roles in both propagating forces and dissipating anisotropic tensions (reviewed in *Godard and Heisenberg, 2019*). During zebrafish epiboly, oriented cell divisions relieve stresses in the EVL to facilitate tissue spreading (*Campinho et al., 2013*). More recently, cell divisions have been shown to contribute to the viscoelastic properties of cells. For example, the deep cells in the central region of the zebrafish blastoderm undergo a rapid decrease in tissue viscosity following mitotic rounding that is important for initiating epiboly (*Petridou et al., 2019*). Similarly, inhibiting cell division in amniote gastrulation resulted in high tissue viscosity, slowing cell Polonaise movements (*Saadaoui et al., 2020*). While our understanding of the effects of cell divisions on tissue development has advanced considerably, the role of the terminal step of cytokinesis, intercellular bridge abscission has not been examined in detail.

Following contraction of the cytokinetic ring, sister cells remain connected via intercellular bridges (*Burgos and Fawcett, 1955*). Actomyosin and microtubule networks constrict the bridges to 1–2 μM (*Mierzwa and Gerlich, 2014*) and subsequently, both cytoskeletal networks are removed from bridges which is followed by bridge cleavage adjacent to the cytokinetic midbody (*Connell et al., 2009*; *Frémont et al., 2017*; *Mierzwa and Gerlich, 2014*). The timing of bridge abscission can affect many biological processes. When abscission is delayed in cancer cell culture models, neighboring cell division events can tear open adjacent cytokinetic bridges resulting in the formation of binucleate cells (*Dambournet et al., 2011*). In mouse ES cells, abscission impacts cell fate by acting as a switch for pluripotency exit (*Chaigne et al., 2020*). Lastly, cytokinetic bridges act as landmarks for lumen development in both cell culture and zebrafish, with the timing of abscission critical for successful lumen expansion (*Rathbun et al., 2020*; *Willenborg et al., 2011*). More specifically, in Kupffer's vesicle, the zebrafish left-right laterality organ, premature cytokinetic bridge cutting via laser ablation disrupted lumen morphogenesis (*Rathbun et al., 2020*). Open questions include what happens to cells if they remain interconnected by cytokinetic bridges during tissue morphogenesis and do cytokinesis failures hinder cell rearrangements or disrupt tissue-scale forces? Furthermore, what subcellular pathways are required for abscission during embryonic development?

Rab proteins are the largest family of small GTPases and are important for intracellular membrane trafficking (*Hutagalung and Novick, 2011*; *Stenmark, 2009*). Endocytic-membrane recycling is critical for both cell division and cytokinetic abscission. In cell culture, Rab11-positive recycling endosomes coordinate cell cycle progression and mitotic spindle positioning (*Hehnly and Doxsey, 2014*). Recycled membrane is trafficked to the midbody and impaired Rab11 or Rab35 endocytic-recycling pathways result in delayed or failed abscission (*Dambournet et al., 2011*; *Frémont and Echard, 2018*; *Frémont et al., 2017*; *Kouranti et al., 2006*). For example, disrupting Rab11 directed trafficking perturbs Kupffer's vesicle morphogenesis during embryonic development in the zebrafish (*Rathbun et al., 2020*).

Here, we characterize Rab25, an epithelial specific membrane recycling protein that is a member of the Rab11 subfamily (*Goldenring et al., 1993*; *Mitra et al., 2017*). Rab25 has been implicated in cancer cell metastasis (*Caswell et al., 2007*; *Mitra et al., 2017*), but its role in embryonic tissue morphogenesis has not been examined. We demonstrate that maternal-zygotic Rab25a and Rab25b mutant embryos exhibit epithelial spreading delays. Here, we propose an EVL trafficking defect in mutant embryos resulted in delayed or failed intercellular bridge abscission during cytokinesis which

slowed epiboly movements. Failure of timely bridge scission resulted in the progressive formation of multinucleate cells which remained mitotically active, resulting in heterogenous cell sizes with anisotropic cell shapes and spatial arrangements in the EVL. Abnormal EVL cells exhibited an overall reduction in cortical actin density during epiboly. Sparse actomyosin networks were associated with uncoordinated cell behaviors, balanced tensions, and altered viscoelastic responses during epiboly.

## Results

### Fluorescently tagged Rab25 localizes near the plasma membrane, centrosomes and cytokinetic midbody

In zebrafish, *rab25a* and *rab25b* transcripts are maternally deposited and their levels increase during gastrulation (https://www.ebi.ac.uk/gxa/home). Whole-mount in situ hybridization revealed ubiquitous expression of both genes in the blastoderm prior to epiboly initiation, followed by EVL restricted expression during epiboly (*Figure 1A*). No expression was detected using sense control probes (*Figure 1—figure supplement 1A*). EVL expression is consistent with reported epithelial-specific function of Rab25 in mammals (*Goldenring et al., 1993*).

To examine the dynamic subcellular distribution of Rab25a and Rab25b in the EVL, N-terminally fluorescently tagged Rab25a and Rab25b constructs were expressed in WT embryos by RNA injection at the one-cell stage. By live confocal microscopy, Venus-Rab25a and eGfp-Rab25b were observed at the plasma membrane and in motile cytoplasmic puncta (*Figure 1B*; *Video 1*). Between dome and 50% epiboly, when the EVL is most proliferative, Venus-Rab25a and eGfp-Rab25b appeared to localize near centrosomes during mitosis and then moved toward the opposing poles of dividing cells (*Figure 1B*, arrowheads; *Video 1*). Injection of *mcherry-rab25b* mRNA into transgenic animals expressing Tubulin-Gfp (*Fei et al., 2019*) confirmed that mCherry-Rab25b localized near centrosomes and dissipated following cell division (*Figure 1C*; arrows). During cytokinetic abscission, co-expression of eGfp-Rab25b with the midbody marker mCherry-Mklp1 demonstrated that eGfp-Rab25b dynamically localized within intercellular bridges adjacent to the midbody (*Figure 1D*), the sites of bridge scission.

Proliferation in the EVL largely subsides during late epiboly progression stages (*Campinho et al., 2013*). Consequently, both eGfp-Rab25b and Venus-Rab25a were primarily distributed near plasma membrane regions between 50 and 80% epiboly, as well as in small dynamic cytoplasmic puncta (*Video 2*). Notably, as marginal EVL cells intercalated into submarginal zones, Rab25 constructs became enriched at tricellular vertices contacting the yolk cell (*Video 3*). Overall, the redistribution of Rab25 constructs to centrosomes and the midbody are suggestive of a role in cell division. While Rab25 has not previously been shown to regulate these processes, the closely related Rab11 has well known functions in mitosis and cytokinesis (*Hehnly and Doxsey, 2014*; *Rathbun et al., 2020*). Furthermore, recruitment of fluorescent-Rab25 to cell vertices is similar to Rab11 distribution in the *Xenopus laevis* neuroepithelium (*Ossipova et al., 2014*), implicating Rab25 in cell shape changes during epiboly.

### Rab25a and Rab25b are required for normal epiboly movements

To explore the functions of Rab25a and Rab25b, CRISPR/Cas9 gene editing was used to generate maternal-zygotic (MZ) mutant lines. Guide RNAs were designed to target exon two which encodes the GTPase domain, the functional domain of Rab proteins (*Mitra et al., 2017*). We characterized two *rab25a* mutant alleles from two founder fish, a 13-base pair (bp) deletion (2.3) and a 24 bp insertion / 3 bp deletion (4). Each allele contained a premature stop codon that disrupted the GTPase domain (see Materials and methods). An 18 bp deletion was generated in *rab25b* which produced an in-frame mutation and deleted a portion of the GTPase domain (see Materials and methods). Quantitative PCR analysis of transcript levels at shield stage showed undetectable levels of *rab25a* transcripts and reduced *rab25b* transcripts in MZ*rab25a* and MZ*rab25b* embryos, respectively (*Figure 2—figure supplement 1A*). Nonsense-mediated decay can result in genetic compensation of similar sequences hence we examined the levels of *rab25a*, *rab25b* and *rab11a* in mutant embryos (*El-Brolosy and Stainier, 2017*; *Rossi et al., 2015*). MZ*rab25a* embryos had elevated levels of *rab25b* transcript, whereas MZ*rab25b* mutants did not have elevated levels of *rab25a* transcript. *rab11a* transcript levels were similar to WT in both mutants. Thus, MZ*rab25a* mutants appeared to

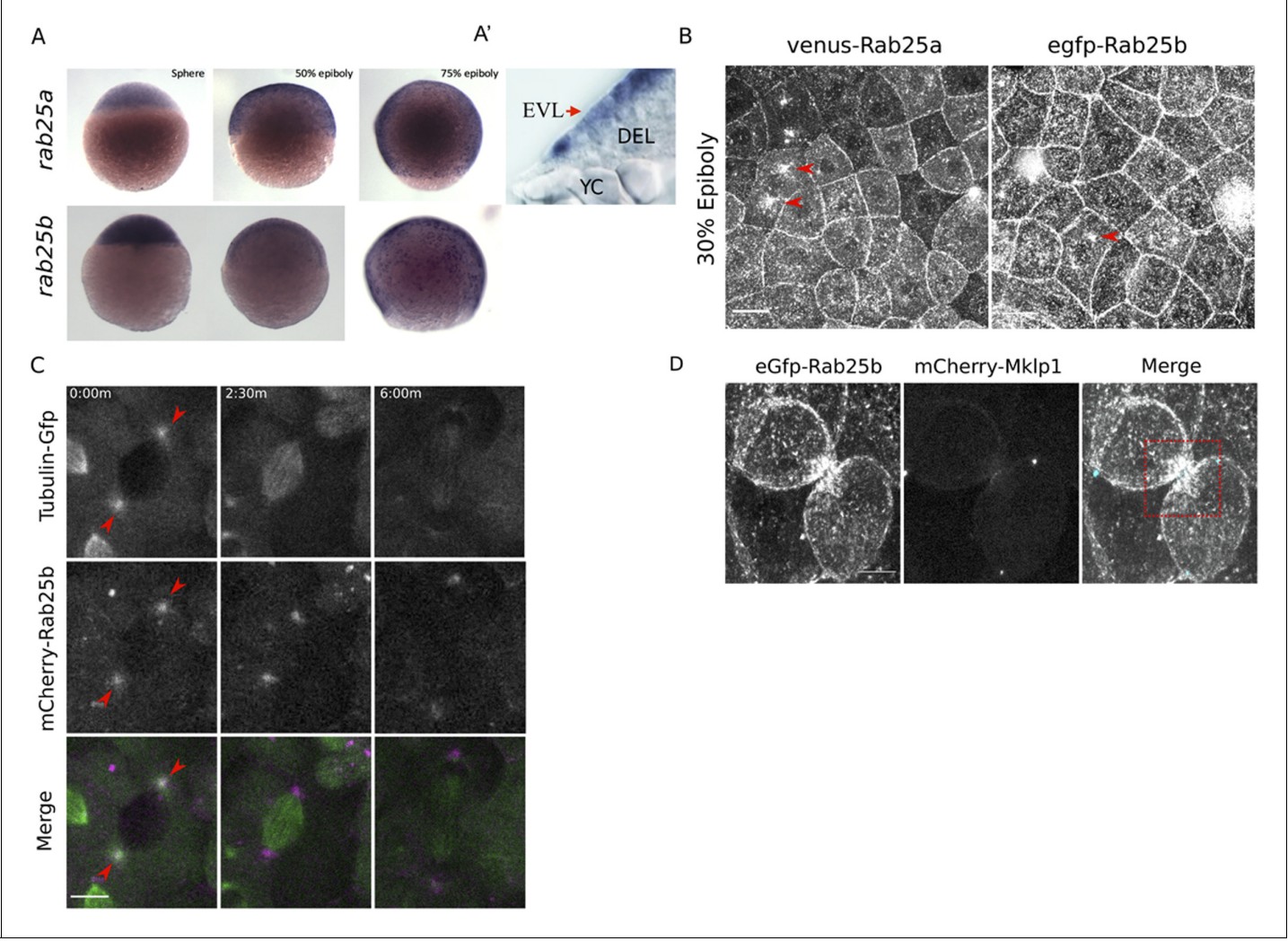

**Figure 1.** *rab25a* and *rab25b* expression pattern and subcellular localization. (**A**) Bright field images of whole-mount in situ hybridizations for *rab25a* (top row) and *rab25b* (bottom row), in WT embryos; lateral views with animal pole positioned to the top. (**A'**) Section of a WT embryo showing *rab25a* expression restricted to the EVL (arrow, top row far right panel); transcripts absent from the deep cells (DEL) and yolk cell (YC). (**B**) Confocal z-projections of Venus-Rab25a and eGfp-Rab25b subcellular localization in WT embryos; red arrowheads denote perinuclear aggregates. Scale bar 20 μm. (**C**) Confocal z-projections of stills from time-lapse movies of transgenic Tubulin-GFP (green) embryos expressing mCherry-Rab25b (magenta). Arrowheads denote co-localization of mCherry-Rab25b at centrosomes. Scale bar 20 μm. (**D**) Confocal z-projections of eGfp-Rab25b (white) and mCherry-Mklp1 (teal) localization in WT embryos; box highlights enrichment of eGfp-Rab25b adjacent to the midbody. Scale bar 20 μm.

The online version of this article includes the following figure supplement(s) for figure 1:

**Figure supplement 1.** Sense probe controls for *rab25a* and *rab25b* in situ hybridizations.

exhibit genetic compensation in the form of transcriptional adaption of *rab25b*, whereas MZ*rab25b* mutants did not. Double MZ*rab25a/rab25b* embryos exhibited phenotypes similar to those of MZ*rab25b* single mutants (*Figure 2—figure supplement 1B*) with moderate increases in the severity of the EVL defects compared to either MZ*rab25a* or MZ*rab25b* single mutants. These results support our hypothesis that genetic compensation by *rab25b* partially rescues MZ*rab25a* mutant embryos, producing a milder phenotype than MZ*rab25b* or MZ*rab25a/rab25b* mutants.

To examine the mutant phenotypes, WT, MZ*rab25a*, and MZ*rab25b* embryos were time-matched and examined live by light microscopy. A subset of MZ*rab25a* (36/100) and MZ*rab25b* (30/100) embryos exhibited abnormal blastoderm morphology during early blastula stages (*Figure 2—figure supplement 1D*). Notably, the majority of MZ*rab25a* (20/36) and MZ*rab25b* (20/30) mutant embryos with early blastoderm defects recovered by 4.3hpf and MZ*rab25a* and MZ*rab25b* embryos initiated

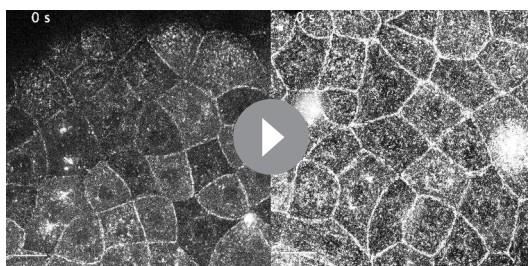

**Video 1.** N-terminally tagged-Rab25 constructs dynamics during cell proliferation and early epiboly. WT embryo expressing Venus-Rab25a (left panel) and eGfp-Rab25b (right panel) beginning at 30% epiboly; Scale bar 20 µm.

https://elifesciences.org/articles/66060#video1

epiboly on time (*Figure 2A*, 4.3hpf, dome). Following this, MZ*rab25a* and MZ*rab25b* mutants exhibited strong epiboly delays that worsened over time, with the greatest delays observed at late epiboly stages (*Figure 2A*, 6-9hpf, *Figure 2B*). The epiboly delay in MZ*rab25b* embryos could be rescued by a transgenic construct expressing full length *rab25b* under the control of a ß-actin promoter, indicating that the phenotype results from the loss of Rab25b (*Figure 2—figure supplement 1C*).

The mutant phenotypes apparently did not result from a general developmental delay, as the zebrafish organizer formed at the same time as time-matched WT embryos (*Figure 2A*, 6hpf, white asterisk). During epiboly, deep cells are positioned behind the leading edge of the EVL and do not move past the EVL margin (reviewed in *Bruce and Heisenberg, 2020*). Examination of deep cell and EVL epiboly revealed they were equally delayed, suggesting the blastoderm delay could be the result of an EVL-specific defect (*Figure 2—figure supplement 1E*). Overall, we found that MZ*rab25b* phenotypes were more severe than either of the MZ*rab25a* alleles. Despite the strong epiboly delay, most MZ*rab25a* (87/100) and MZ*rab25b* (99/100) embryos completed gastrulation and survived to adulthood.

We examined patterning and germ layer specification by in situ hybridization. EVL differentiation is required for normal epiboly (*Fukazawa et al., 2010*), and we found that expression of the EVL marker *keratin4* (*krt4*) was similar in mutant and WT embryos (*Figure 2—figure supplement 1F*). Analysis of the mesoderm marker, *goosecoid*, and the endodermal marker *sox17*, also showed normal expression patterns (*Figure 2—figure supplement 1F*). Occasionally, *sox17* staining showed that the dorsal forerunner cluster was disorganized in MZ*rab25a* and MZ*rab25b* embryos (*Figure 2—figure supplement 1F*). Overall, germ layer specification appeared to be largely normally in mutant

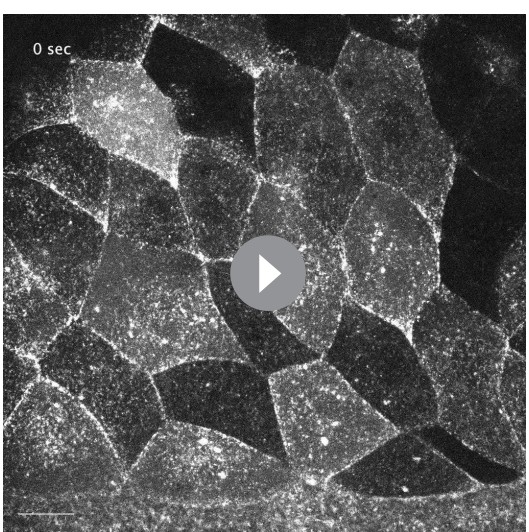

**Video 2.** eGfp-Rab25b live distribution at the plasma membrane and cell vertices during late epiboly. WT embryo expressing eGfp-Rab25b beginning at 6hpf; Scale bar 20 µm.

https://elifesciences.org/articles/66060#video2

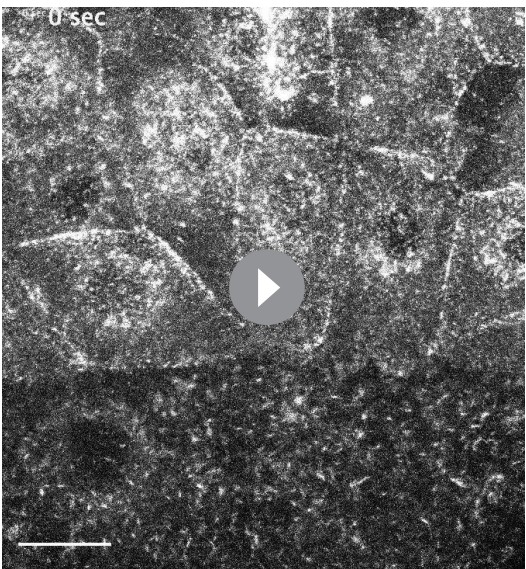

**Video 3.** Venus-Rab25a is dynamically recruited to vertices of rearranging marginal cells. WT embryo expressing Venus-Rab25a beginning at 7hpf; Scale bar 20 µm.

https://elifesciences.org/articles/66060#video3

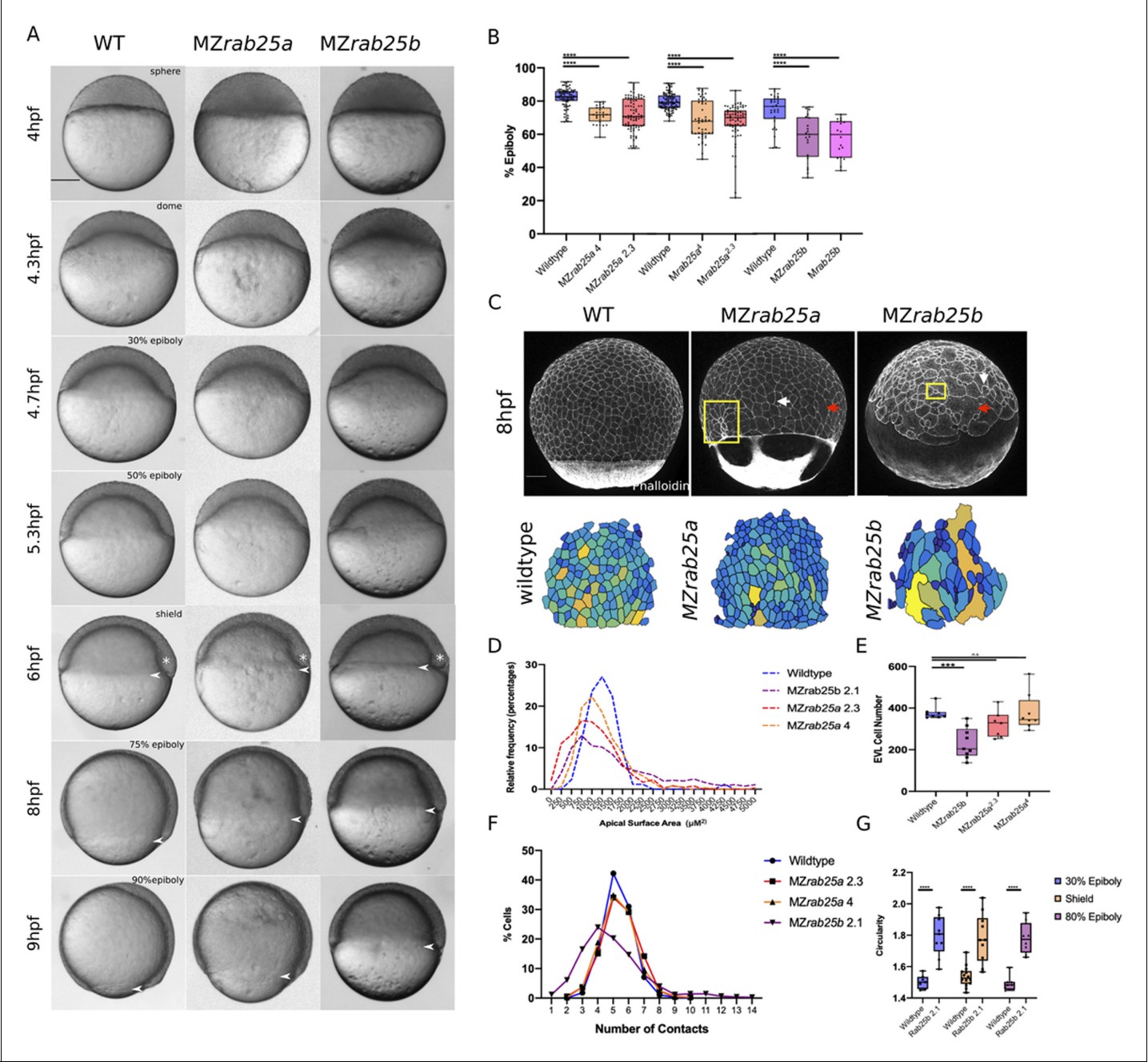

**Figure 2.** Epithelial spreading delays and heterogenous cell morphology, size and spatial arrangements in MZ*rab25a* and MZ*rab25b* embryos. (**A**) Time-matched bright field images of lateral views of WT, MZ*rab25a* and MZ*rab25b* embryos during epiboly. Arrowheads indicate blastoderm margin, asterisks denote embryonic organizer (shield). (**B**) Quantification of epiboly progression after 8hpf in: WT (n = 80), MZ*rab25a* (2.3, n = 87), MZ*rab25a* (4,n = 28); WT (n = 97), M*rab25a* (2.3,n = 73), MZ*rab25a* (4,n = 49); WT (n = 29), MZ*rab25b* (n = 21), M*rab25b* (n = 18). Means: SEM; Two-Way ANOVA; ***p<0.001, ****p<0.0001.(N = 3). (**C**) Confocal z-projections of time-matched lateral views of WT, MZ*rab25a* and MZ*rab25b* embryos at 8hpf stained with phalloidin and corresponding apical surface area heat maps. Cooler colors represent smaller areas, warmer colors represent larger areas. Yellow boxes indicate cells with reduced apices. Red arrows denote cells with increased apical surface areas. White arrows indicate curved cell junctions. Scale bar 100 µm. (**D**) Frequency distribution of apical surface areas of WT (n = 817, N = 8), MZ*rab25a* 2.3 (n = 651, N = 14), MZ*rab25a* 4 (n = 654, N = 15) and MZ*rab25b* (n = 503, N = 15) embryos at 6hpf. (**E**) EVL Cell number in WT (n = 8), MZ*rab25b* (n = 8), MZ*rab25a* 2.3 (n = 9) and MZ*rab25a* 4 (n = 7) embryos at 6hpf. Means: SEM; Two-Way ANOVA; ***p<0.001. (**F**) Frequency distributions of EVL cellular contacts number at 6hpf in WT (n = 817, N = 8), MZ*rab25a* 2.3 (n = 651, N = 14), MZ*rab25a* 4 (n = 654, N = 15) and MZ*rab25b* (n = 503, N = 15). (**G**) Circularity quantifications for WT and MZ*rab25b* embryos during epiboly. 30% epiboly: WT (n = 7), M*rab25b* (n=7). Shield: WT (n=14), M*rab25b* (n=10). 80% epiboly: WT (n=8), M*rab25b* (n=9). Means: SEM: One-way ANOVA, ****p<0.0001.

*Figure 2 continued on next page*

*Figure 2 continued*

The online version of this article includes the following figure supplement(s) for figure 2:

**Figure supplement 1.** MZ*rab25a* and MZ*rab25b* phenotypic characterization.
**Figure supplement 2.** MZ*rab25a* and MZ*rab25b* phenotypic characterization.

embryos. Together these data suggested that Rab25a and Rab25b may have a specific role in epiboly during zebrafish gastrulation.

## Cell shape and rearrangements indicative of epithelial defects in MZ*rab25*a and MZ*rab25*b embryos

The EVL restricted expression of *rab25a* and *rab25b* pointed to a primary defect in the EVL of Rab25 mutant embryos, consistent with the epithelial-specific role of Rab25 in mammals (*Goldenring et al., 1993*; *Jeong et al., 2019*). Consistent with an EVL defect, analysis of phalloidin stained Rab25 mutant embryos revealed striking cellular size and shape heterogeneity compared to wild type (*Figure 2C*). Our examination also revealed that yolk cell actin networks were perturbed in MZ*rab25a* and MZ*rab25b* embryos, suggesting yolk cell defects may contribute to the mutant phenotypes (*Figure 2C*). Disorganization of the yolk cell cytoskeleton can be the result of defective egg development and can impact the rate of epiboly (*Li-Villarreal et al., 2016*). Thus, we examined the cortical actin networks of four-cell stage mutant embryos by phalloidin staining to address this possibility.

While WT (8/8) and MZ*rab25a* (10/10) yolk cell actin networks were comparable, a small number of MZ*rab25b* (1/12) embryos displayed localized yolk cell cortical actin disorganization (*Figure 2—figure supplement 2A*). Thus, while we cannot exclude the possibility that actin defects in the egg contributed to the yolk cell phenotypes, it seemed unlikely that they produced the morphological defects in mutant EVL cells. In support of this, EVL cell morphology was normal at the onset of epiboly following cleavage stages in all mutant embryos examined (*Figure 2—figure supplement 2C*). By shield stage, mutant EVL cells exhibited a range of defects compared to wild type (*Figure 2—figure supplement 2C*). Furthermore, these defects preceded the greatest epiboly delays which occurred between 6-8hpf, suggesting a relationship between the EVL defects and the epiboly delays. Lastly, depolymerizing actin in the yolk cell of wild-type embryos failed to cause EVL defects that resembled those in Rab25 mutant embryos (*Figure 2—figure supplement 2B,B'*). Taken all together, these findings suggest that the EVL phenotypes in Rab25 mutant embryos are likely to be cell autonomous. Therefore, we focused our analysis on the EVL of MZ*rab25a* and MZ*rab25b* mutants during epiboly.

Stage-matched embryos at 60% epiboly were used to quantify the EVL defects. This stage was chosen because that was how far epiboly progressed in most mutants by 8hpf and it was also when embryos exhibited the largest delays (*Figure 2A,B*). EVL mean surface area measurements, heat maps, and frequency distributions were generated to assess the surface areas of cells in phalloidin stained mutant and WT embryos. MZ*rab25a* and MZ*rab25b* embryos exhibited heterogenous cell surface areas (*Figure 2C,D*). In WT, EVL cells had apical surface areas between 750 and 1750 µm, while the distribution of EVL surface areas in MZ*rab25a* and MZ*rab25b* mutant embryos was broader, indicating both smaller and larger cell apices (*Figure 2D*).

Heterogeneity of apical surface area in mutants was associated with anisotropic cell shapes and disorganized spatial arrangements. While WT cells were pentagonal or hexagonal, as expected, MZ*rab25a* and MZ*rab25b* cells exhibited cell shape deviations and abnormal numbers of neighbors (*Figure 2F,G*). This was reflected by circularity measurements and cell-cell contact number, with larger circularity values indicating increased cell shape anisotropy (*Figure 2G*). Cells with increased apical surface area tended to have increased junctional curvature (*Figure 2C*, red arrows), whereas cells with reduced apices tended to be round (*Figure 2C*, yellow boxes). Notably, normally sized cells in mutants still displayed anisotropic cell shapes and non-linear cell contacts (*Figure 2C*, white arrows). Compared to WT embryos, the EVL defects were associated with reduced cell number in MZ*rab25b* but not MZ*rab25a* embryos (*Figure 2E*).

Cell size is reported to scale with nuclear number (*Cao et al., 2017*). Thus, we examined whether mutant cells with large apical surface areas were multinuclear by labeling WT, MZ*rab25a* and

MZ*rab25b* embryos with membrane-Gfp and the nuclear marker H2A-Rfp. Large cells in MZ*rab25a* and MZ*rab25b* embryos were bi- or multinucleate (*Figure 3A*), thus cell size scaled with nuclear number (*Figure 3B*). Overall, multinucleate large cells suggested cytokinesis failures in MZ*rab25a* and MZ*rab25b* embryos (*Rathbun et al., 2020*), consistent with the emergence of EVL defects following epiboly initiation, when the EVL is most proliferative (*Campinho et al., 2013*).

## Cytokinetic abscission defects during epiboly in MZ*rab25b* mutants

Localization of eGfp-Rab25b near the cytokinetic midbody and the presence of large multinuclear EVL cells in MZ*rab25a* and MZ*rab25b* embryos indicated that Rab25 may function in cytokinesis. To investigate potential abscission defects in Rab25 mutants, we focused on MZ*rab25b* embryos because they exhibited more severe defects than MZ*rab25a* embryos. Cytokinetic bridges were characterized by labeling microtubules, nuclei and membrane in WT and mutant embryos. Following mitosis in WT cells, equatorial midzone microtubules were organized into apical cytokinetic bridges (*Figure 3C*, white arrows, *Video 4*). Initially, cytokinetic bridges underwent a fast phase of bridge narrowing and shortening, over approximately 10–12 min (*Figure 3C*, t = 16:36 m-27:40 m). Intercellular bridges then remained connected for an additional 15–20 min, in which bridge length did not change significantly before abscission (*Figure 3C*, t = 27:40-38:48 min).

During mononucleate cytokinesis in MZ*rab25b* embryos, the plasma membrane at the cleavage furrow underwent ingression similar to WT embryos (*Figure 3C', C"*). Following cleavage, two types of abscission defects were detected. The first occurred during the initial, fast phase of bridge formation when bridge regression occurred in a small proportion of mutant cells during cytokinesis (5/23), leading to the formation of binucleate cells (*Figure 3C'*, yellow arrows, *Video 5*). In these instances, MZ*rab25b* embryos expressing the midbody marker mCherry-Mklp1, showed a failure of mCherry-Mklp1 coalescence in unstable cytokinetic bridges, which preceded bridge regression (*Video 6*; WT (left panel), MZ*rab25b* (right panel)).

The second defect occurred later during the abscission process. Despite being initially longer following their formation (*Figure 3E*), cytokinetic bridges in these contexts appeared to shorten and narrow normally (*Figure 3C''*, red arrowhead). In WT embryos, daughter cells remained interconnected for ~26.23 min before bridge scission (n = 17), while in mutants, bridges persisted almost twice as long, suggesting abscission either failed or was delayed (*Figure 3D*) (~50.35 mins; n = 15). While some bridges were still present at the end of the time-lapses, most long-lasting intercellular bridges appeared to be pulled open by morphogenetic stress in adjacent EVL regions, such as mitosis or basal cell extrusion, resulting in multinucleation (*Figure 3C"*, red arrowheads, *Video 7*). Overall, these observations are similar to in vitro work showing intracellular membrane trafficking regulates both cytokinetic bridge stability and timing of abscission (*Dambournet et al., 2011*).

## Multipolar cytokinesis in MZ*rab25b* embryos

Binucleate cells remained mitotically active in MZ*rab25b* embryos, resulting in continuous cytokinesis failures which produced multinucleated cells. Notably, in MZ*rab25b* embryos, spindles in mono- and multinucleated EVL cells deviated from both the long axis of the cell and the animal-vegetal axis (*Figure 4—figure supplement 1A,A'*). Thus, misoriented spindles likely contributed to the disorganized EVL cell arrangements (*Campinho et al., 2013*). During multipolar cytokinesis, both cleavage and abscission failures were seen. In some multinucleated cells, the plasma membrane appeared to snap back following furrow ingression, suggesting that the F-actin cytokinetic ring might be unstable (6/21) (*Figure 4—figure supplement 1D*). To investigate actin dynamics during cytokinesis, embryos were co-injected with RNAs encoding membrane-Rfp and Gfp-Utrophin. Cytokinetic rings progressed from basal to apical, as in other vertebrate epithelia (*Figure 4A,B*; *Higashi et al., 2016*). During multinucleate cell cleavage in mutant embryos, we often observed failed contraction of the actin cytokinetic ring of EVL cells followed by cytokinesis failures (*Figure 4C*).

When multinucleate cells in MZ*rab25b* mutants successfully completed cleavage (15/21), multiple daughter cells were generated that typically had reduced apical surface areas compared to surrounding cells (*Figure 4—figure supplement 1D*). Thus, successful cleavage during multipolar cytokinesis likely explains the presence of cells with reduced apical surface areas in mutant embryos. Most, but not all, daughter cells contained nuclei (not shown). Multipolar cytokinesis resulted in arrays of daughter cells interconnected through cytokinetic bridges across the EVL in MZ*rab25b*

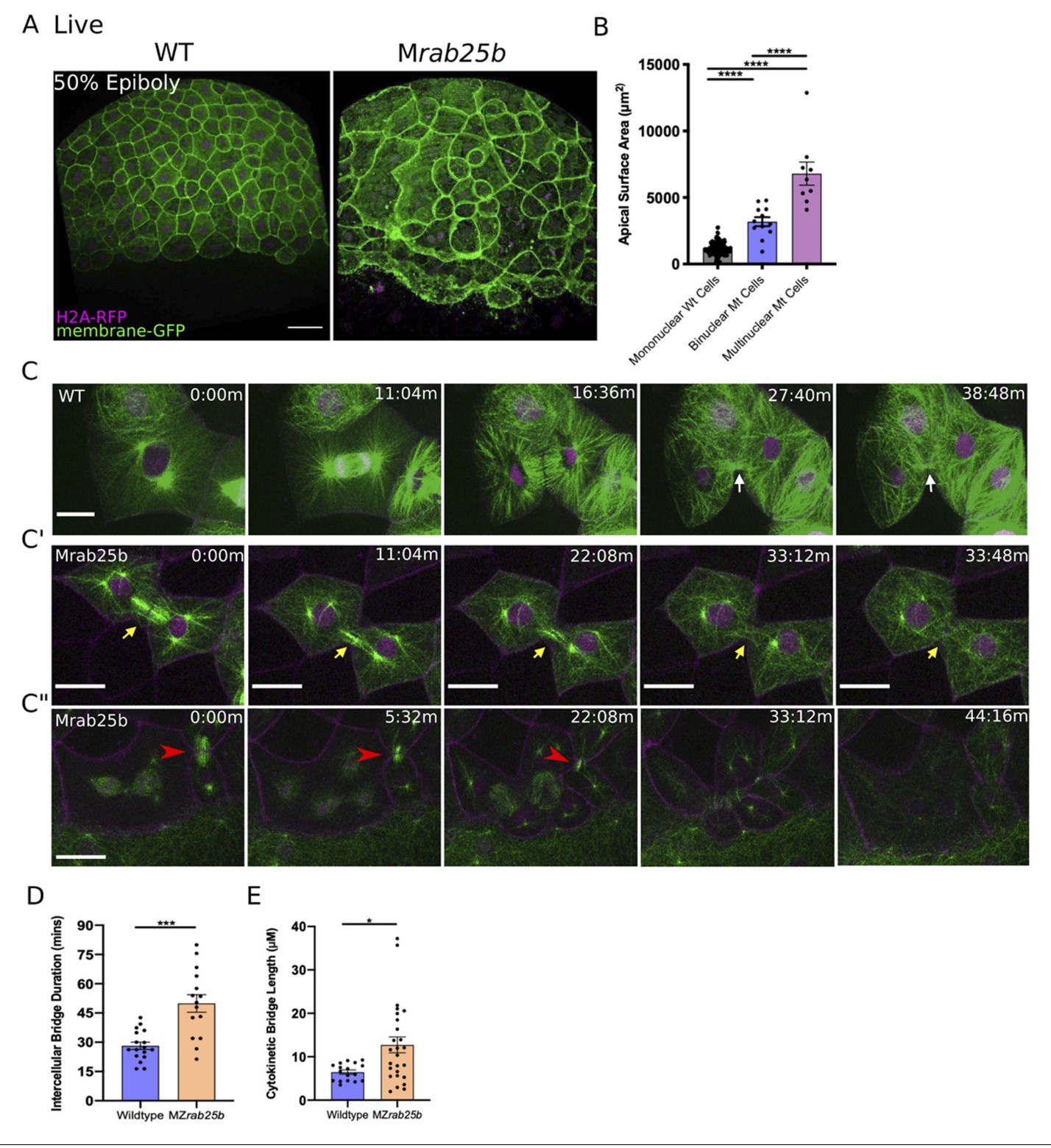

**Figure 3.** Unstable cytokinetic bridges and delayed abscission in MZ*rab25b* embryos. (**A**) Confocal z-projections of stage-matched WT and MZ*rab25b* embryos at 50% epiboly expressing mGfp (green) and H2A-Rfp (magenta). Embryos positioned laterally. Scale bar 40 μm. (**B**) Apical surface area quantifications of WT mononucleate (n = 151, N = 6), MZ*rab25a* binucleate (n = 10, N=6) and MZ*rab25b* bi-(n = 3, N=4) and multinucleate (n = 9, N=4) at 80% epiboly. Mean: SEM; One-Way ANOVA; ****, p<0.0001. (**C–C”**) z-projection of stills from confocal time-lapses of WT and MZ*rab25b* embryos labeled for microtubules (green), nuclei (magenta), and plasma membrane (magenta) starting at dome stage; white arrow marks WT bridge; yellow

*Figure 3 continued on next page*

*Figure 3 continued*

arrows marks bridge regression in MZ*rab25b* cell. (**C"**) Red arrowheads indicate cytokinetic bridge torn open by neighboring morphogenic stress. Scale bar 20 μm. (**D**) Intercellular bridge duration in WT (n = 17, N = 3) and MZ*rab25b* (n = 15, N = 4) embryos during epiboly initiation in EVL marginal regions. Mean: SEM; Mann- Whitney Test; ***, p<0.001. (**E**) Intercellular bridge length following formation in WT (n = 17,N = 3) and MZ*rab25b* (n = 26, N = 4) embryos during epiboly initiation in EVL marginal regions. Mean: SEM; Mann-Whitney Test; *p<0.05.

embryos (*Figure 4D*, numerals and arrowheads, *Video 8*). As seen in mononuclear cell divisions, in multipolar divisions, intercellular bridges were not observed to undergo abscission but instead appeared to be torn open due to morphogenic stress, causing cytokinesis failures (*Video 8*). Following multipolar division failures, Myosin-Gfp foci were often observed along individual cell junctions, suggesting myosin was unevenly distributed along these EVL cell contacts (*Figure 4C'*, white arrows). Myosin-Gfp foci were apparently remnants of de novo tricellular vertices that are normally established along newly formed daughter cell interfaces, as observed in *Xenopus* during ectoderm cell divisions (*Higashi et al., 2016*).

We observed that cytokinetic intercellular bridge abscission failures resulted in the progressive formation of bi-and multinucleate cells during epiboly in MZ*rab25b* embryos. The abscission defects disrupted EVL organization in mutant embryos, as cells exhibited variable sizes, shapes, and number. Contributing to the tissue defects were basal cell extrusion events, cell-cell fusions and misoriented mitotic spindles (*Figure 4—figure supplement 1E,F*). Cytokinesis defects causing epithelial spreading delays aligns with previous work implicating cell division in epithelial cell rearrangements and epiboly (*Campinho et al., 2013*; *Higashi et al., 2016*).

## Marginal EVL cell rearrangements are disrupted during epiboly progression in MZ*rab25b* embryos

The cellular defects in MZ*rab25a* and MZ*rab25b* embryos appeared largely to be the product of cytokinesis failures. We next sought to characterize cell behaviors during later epiboly stages, when the epiboly delay was greatest. During epiboly progression, marginal EVL cells elongate along the animal-vegetal axis and a subset of cells rearrange and intercalate into submarginal regions which enables the marginal circumference to narrow to eventually close the blastopore (*Keller and Trinkaus, 1987*; *Köppen et al., 2006*). To analyze EVL cell shape changes, live imaging of wild type and mutant embryos expressing mRfp and Gfp-Utrophin was performed starting at 7 hpf.

In wild-type embryos, marginal EVL cells elongated as expected and a few cells changed shape by shortening their contacts with the yolk cell along the EVL margin (*Figure 5A*, green cells). Following these shape changes, three to four cells share one vertex with the underlying yolk cell (*Figure 5A*, red circle, t = 25:26 m), akin to a multicellular rosette. Resolution of rosettes in other contexts is linked to cell intercalation rates and tissue-shape changes (*Blankenship et al., 2006*; *Zallen and Blankenship, 2008*). Multicellular-yolk vertices resolved on average 22 min following their formation (*Figure 5E*). Resolution of marginal EVL rosettes resulted in de novo junctions formed between cells adjacent to intercalating cells as they exited into submarginal EVL regions (*Figure 5A*, orange cells, dotted line). Enrichment of cortical actin was observed when EVL cells intercalated into submarginal zones (*Figure 5A'*, *Video 9*, left panel).

Overall, EVL-YC junction shortening took longer in MZ*rab25b* embryos compared to wild-type embryos (*Figure 5C*). Additionally, while WT EVL-YC contact shrink rates were initially fast and slowed over time, MZ*rab25b* contacts displayed more uniform, slower rates of shortening during epiboly (*Figure 5D*). Cells of all sizes were observed to shorten their yolk cell contacts in mutant embryos (*Video 9*, right panel). However, compared to wild-type, mutant cells with normal and large apical surface areas were slower to shrink their yolk cell contact, while cells

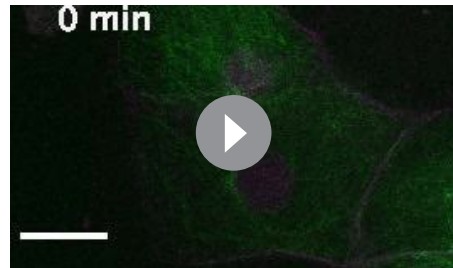

**Video 4.** WT EVL Mitosis. WT embryo injected with *mrfp* (magenta) and *gal4* mRNA and *H2A-rfp*(magenta)-dUAS-*dcx-gfp* (green) plasmid imaged beginning at dome stage. Scale bar 20 μm.
https://elifesciences.org/articles/66060#video4

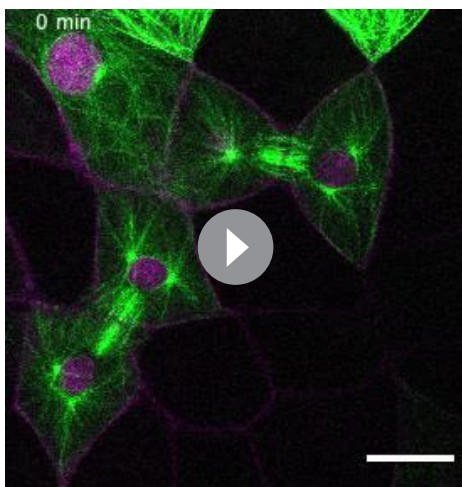

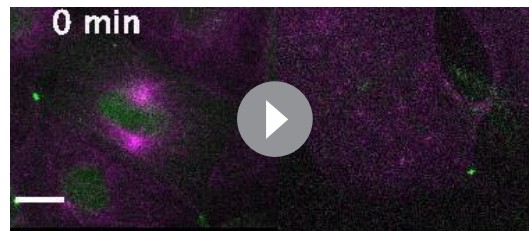

**Video 5.** Cytokinetic bridge regression in M*rab25b* mutant. M*rab25b* embryo injected with *mrfp (magenta)* and *gal4* mRNA and *H2A-rfp*(magenta)-dUAS-*dcx-gfp* (green) plasmid imaged beginning at dome stage. Scale bar 20 µm.

https://elifesciences.org/articles/66060#video5

**Video 6.** Failed formation of the cytokinetic midbody in MZ*rab25b* embryos. Left panel: Tg:(Dcx-Gfp) (magenta) expressing mCherry-Mklp1 (green) imaged beginning at dome stage showing formation of midbody in wild-type embryo. Right panel: Apical z-projection of M*rab25b* embryo expressing mRfp (magenta) and mCherry-Mklp1 (green); coalescence of the cytokinetic midbody which precedes bridge regression; Scale bar 20 µm.

https://elifesciences.org/articles/66060#video6

with reduced apices did so more quickly (*Figure 5—figure supplement 1A*). This data suggests that the cell size defects impacted the rate of EVL cell shape changes, contributing to the slowed rate of epiboly in mutant embryos.

Resolution of multicellular-yolk vertices was significantly slower in mutant embryos compared to wild type (*Figure 5E*). A large proportion of rosettes in marginal regions persisted for up to 2 hr following their formation (*Figure 5E*, red data points), preventing timely intercalation of cells into sub-marginal zones and potentially slowing overall EVL marginal circumference narrowing. Additionally, marginal rosettes in MZ*rab25b* embryos appeared to deform the EVL-YC boundary, causing the margin to buckle animally, resulting in an uneven EVL margin (*Figure 5—figure supplement 1B*) and potentially hindering epiboly progression. Given that normal sized cells exhibit both slow vertex resolution and intercalation rates compared to wild type and that tagged-Rab25 constructs localized to vertices during marginal EVL rearrangement events (*Video 3*), suggests that Rab25 may have an additional direct role in promoting EVL cell intercalation events during epiboly.

Despite the epiboly delay in MZ*rab25b* embryos, EVL cells were able to rearrange into submarginal zones (*Figure 5B*, purple cells), with adjacent cells with large apices being stretched toward the site of intercalation (*Figure 5B'*, colored cells). Marginal EVL cell stretching was in contrast to wild-type cell shape changes and would be predicted to slow the rate of epiboly, as cells normally maintain their shape or narrow and elongate along the embryonic AV axis (*Köppen et al., 2006*).

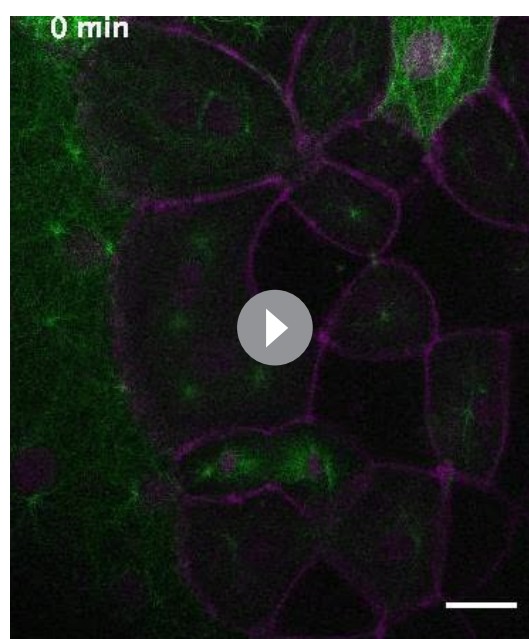

**Video 7.** Torn open intercellular bridge during multipolar mitosis. M*rab25b* embryo injected with *mrfp (magenta)* and *gal4* mRNA and *H2A-rfp* (magenta)-dUAS-*dcx-gfp* (green) plasmids imaged beginning at dome stage. Scale bar 20 µm.

https://elifesciences.org/articles/66060#video7

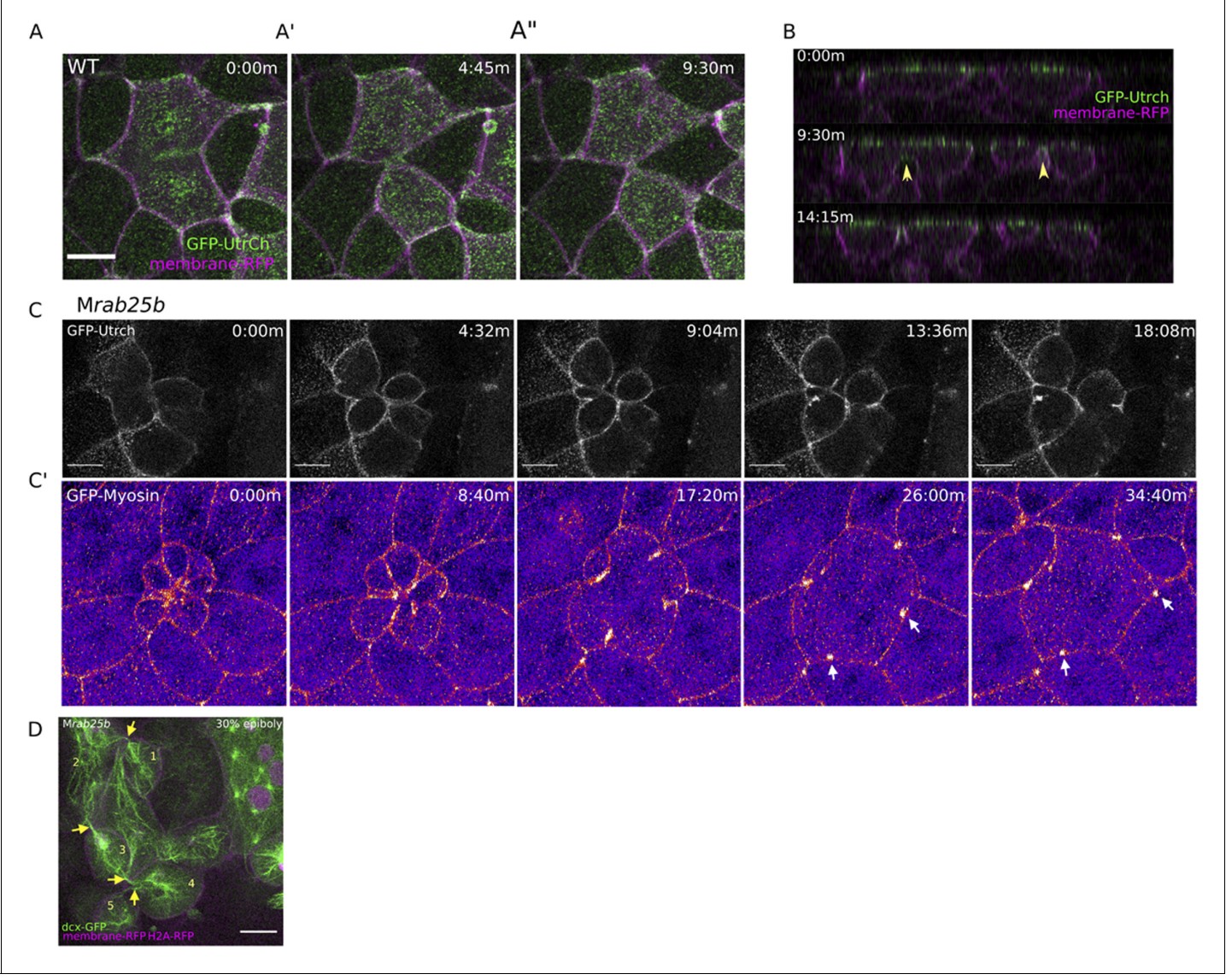

**Figure 4.** Multipolar cytokinesis failure in MZ*rab25b* embryos. (A–A") Confocal z-projections of stills from time-lapse of a WT EVL cell during mitosis expressing Gfp-Utrophin (green) and mRfp (magenta). Scale bar 20 μm. (B) Lateral views with apical to the top of stills from single-plane confocal time-lapses of WT EVL cells during mitosis expressing Gfp-Utrophin (green) and mRfp (magenta). Arrowheads denote cleavage furrow ingression from basal to apical. (C) Confocal z-projections of stills from time-lapse of MZ*rab25b* multipolar cleavage failure. F-actin labeled with Gfp-Utrophin. Scale bar 20 μm. (C') Confocal z-projections of time-lapse of MZ*rab25b* Tg (Myl1.1-Gfp) (Fire-LUT) during multipolar cytokinesis failure. White arrows indicate Myosin-Gfp foci. (D) Confocal z-projection of MZ*rab25b* embryo showing an array of EVL cells interconnected via cytokinetic bridges at 30% epiboly. Microtubules (green), nuclei (magenta), and plasma membrane (magenta), arrows and numbers denote connected cells and cytokinetic bridges. Scale bar 20 μm.

The online version of this article includes the following figure supplement(s) for figure 4:

**Figure supplement 1.** Cytokinesis-related defects in MZ*rab25b* embryos.

## Actomyosin network organization in MZ*rab25b* embryos

Our observations that EVL cells in mutant embryos exhibit misoriented and uncoordinated cell shape changes prompted us to examine F-actin and myosin in more detail, given their fundamental roles in cell shape and rearrangements (*Jewett et al., 2017*). A gradual reduction in cortical actin over the course of epiboly was observed in phalloidin stained mutant embryos, which was first detected at 30% epiboly in MZ*rab25b* embryos (*Figure 6A–A'*, *Figure 6—figure supplement 1A*). While cells of

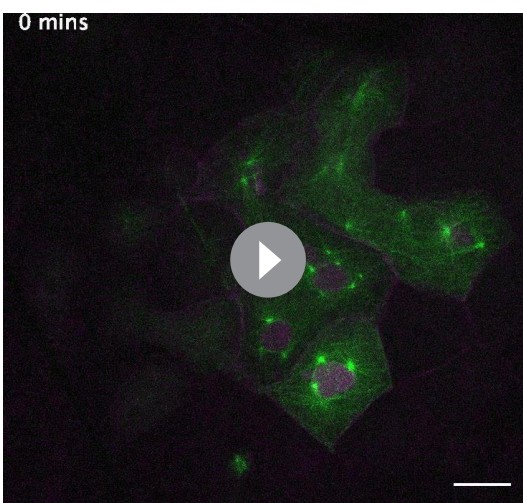

**Video 8.** EVL cells interconnected by long cytokinetic bridges in M*rab25b* embryos. *Mrab25b* embryo injected with *mrfp (magenta)* and *gal4* mRNA and *H2A-rfp*(magenta)-dUAS-*dcx-gfp (green)* plasmid imaged beginning at dome stage. Scale bar 20 μm.
https://elifesciences.org/articles/66060#video8

all sizes exhibited reduced F-actin signal intensity (*Figure 6A*, red arrows, *Figure 6—figure supplement 1B*), as cell size increased, cortical actin further decreased (*Figure 6A*, yellow arrows).

Antibody staining for phosphorylated myosin (pMyosin), the active form of myosin and a proxy for contractility, revealed that most EVL cells in MZ*rab25b* embryos exhibited reduced junctional pMyosin compared to wild-type embryos at 80% epiboly (*Figure 6B,B'*). In MZ*rab25b* embryos, we occasionally observed some EVL cells with elevated levels of cortical pMyosin (*Figure 6B"*). Furthermore, while junctional pMyosin was uniformly distributed along individual cell-cell contacts in wild-type embryos (*Figure 6A*), in MZ*rab25b* mutants, it was often diffuse and fragmented along cell-cell contacts (*Figure 6B'–B"*, arrows). Overall, reduced cortical actin was associated with weak junctional pMyosin in MZ*rab25b* embryos, with some localized increases in pMyosin in individual cells.

We next characterized the distribution of actin and myosin in live embryos. In MZ*rab25b* embryos transgenic for Myosin light chain 12, genome duplicate 1-gfp (Myl12.1-Gfp) myosin intensity was often weak at shield stage compared to wild type and became heterogeneously distributed in the EVL, as some cells contained elevated levels of cortical myosin following aberrant shape changes (*Video 10*). Gfp-Utrophin was also disorganized in MZ*rab25b* EVL cells following aberrant cell behaviors (*Video 9*, right panel). Collectively, these data indicated that ectopic activation of actomyosin in localized EVL regions are likely the result of disorderly cell movements and rearrangements. This is similar to the accumulation of actomyosin in the *Drosophila* germband during unorganized cell rearrangements (*West et al., 2017*). We propose the abnormal cell shape changes and behaviors in MZ*rab25b* mutant embryos are likely the product of the overall reduction of contractile actomyosin networks and disrupted tissue architecture.

## Altered viscoelastic responses in MZ*rab25a* mutant EVL cells

The initial recoil velocity of laser ablated junctions is a measure of tension (*Hutson et al., 2003*). Reduced cortical actin and pMyosin suggested decreased contractility in the EVL of MZ*rab25a* and MZ*rab25b* embryos. To examine this, junctions perpendicular and parallel to the EVL-YSL margin were ablated in WT and MZ*rab25a* mutant embryos that were transgenic for Myosin-Gfp. We analyzed cells near the EVL margin in embryos at 60–70% epiboly, when EVL tension is highest (*Campinho et al., 2013*) and epiboly was the most delayed in MZ*rab25a* and MZ*rab25b* embryos. To eliminate potential effects of contact length on tension, only cell junctions with similar lengths were quantified.

In WT embryos, cell vertices at the ends of contacts parallel to the margin recoiled faster than perpendicular vertices (*Figure 6C,C', E*). Cells behave as viscoelastic materials; thus, tissue viscosity can impact junctional recoil velocities. Laser ablation results can be modeled as the damped recoil of an elastic fibre, using a Kelvin-Voigt element to represent the junction (*Kumar et al., 2006*). In this context, the maximum distance retracted after ablation is proportional to the stress sustained by the junction, while the rate at which that maximum distance is attained (measured by a relaxation time) is proportional to the viscoelastic properties of the tissue. The relaxation times of parallel and perpendicular junctions following ablation were not significantly different, suggesting that the difference in initial recoil velocities can be attributed to differences in tension, not viscosity (*Figure 6F*). Stress modeling was consistent with the initial recoil velocities (*Figure 6G*). Elongating junctions perpendicular to the yolk cell having slow recoil velocity is similar to tensions reported for fluidizing, growing junctions during wound healing (*Tetley et al., 2019*). Overall, our data suggests that as the

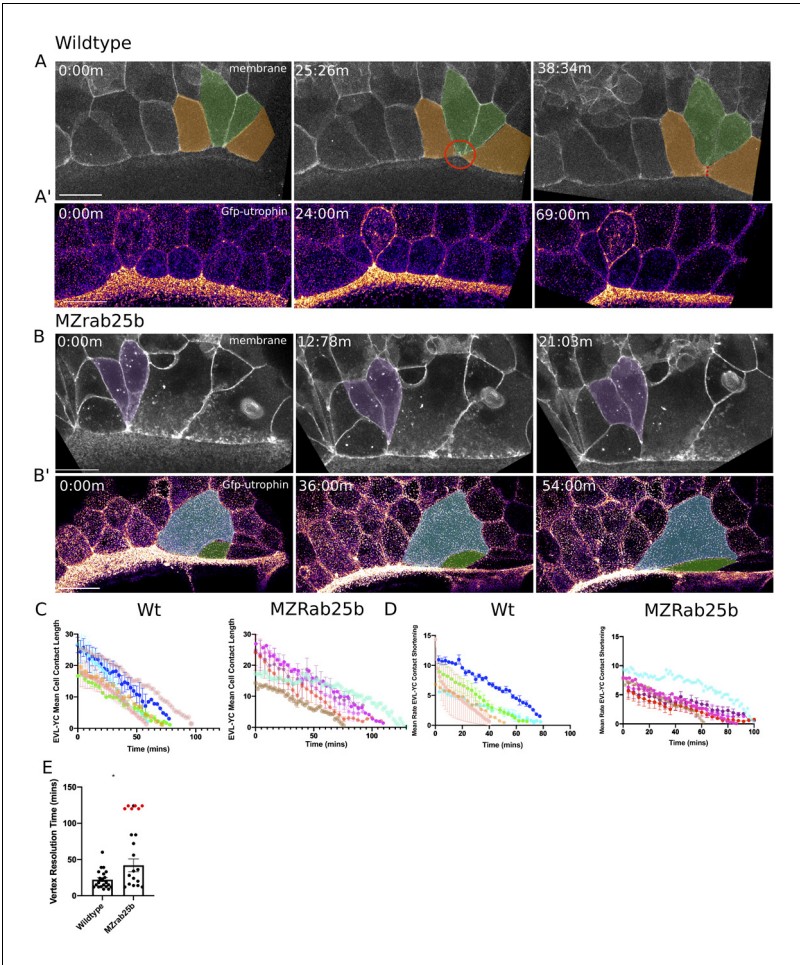

**Figure 5.** Aberrant EVL cell rearrangements in MZ*rab25b* embryos. (**A**) Confocal z-projections of stills from a time-lapse of a WT embryo expressing mGfp starting at 7hpf. Lateral view focused on the margin. Green shaded cells shrink the EVL-YC junction and intercalate into submarginal zones. Orange shaded cells establish new cell-cell contacts following intercalation events (denoted by red dotted line). Circle denotes shared vertex with underlying yolk cell. Scale bar 20 μm (**A'**) Confocal z-projection time-lapse of WT embryo labeled for F-actin; (Fire-LUT) (Gfp-Utrophin) starting at 7hpf; lateral view; scale bar 20 μm. (**B**) Confocal z-projection of stills from a confocal time-lapse starting at 7hpf of an MZ*rab25b* embryo expressing membrane-Gfp. Purple shaded cells exit EVL marginal region. Scale bar 20 μm (**B'**) Confocal z-projection of stills from time-lapse starting at 7hpf of an MZ*rab25b* embryo labeled for F-actin (Fire-LUT) (Gfp-Utrophin); scale bar 20 μm. Shaded cells denote an EVL circumferential stretching event. Scale bar 20 μm. (**C–D**) EVL-YC mean contact length or shortening rate over time in rearranging EVL marginal cells in WT (N = 5) and MZ*rab25b* embryos (N = 5). Mean:SEM. Each color indicates a separate trial of a single embryo. Each line represents the average of the contact length or junction shrink rate in each trial (n = 2–5). (**E**) Resolution times following formation of EVL-YC multicellular vertices. Mean: SEM. WT (n = 20,N = 4) and MZ*rab25b* (n = 12,N = 5), unresolved MZ*rab25b* vertices (red) (n = 6,N = 5). Mann-Whitney, *p<0.05.

The online version of this article includes the following figure supplement(s) for figure 5:

**Figure supplement 1.** Marginal EVL cell behaviors during epiboly progression.

yolk-cell begins pulling the EVL between 60 and 75% epiboly, mechanical stresses are anisotropic, being highest along cell contacts oriented parallel to the EVL-YC boundary in marginal EVL regions.

In MZ*rab25a* embryos, the initial recoil velocities of parallel and perpendicular junctions were significantly different, but this difference was diminished compared to wild-type embryos (***Figure 6D–E***). Additionally, both parallel and perpendicular junctions in MZ*rab25a* mutants exhibited slower recoil velocities compared to wild type (***Figure 6E***). Stress modeling was consistent with both these results (***Figure 6G'***). However, the relaxation time of the EVL following the initial recoil was much

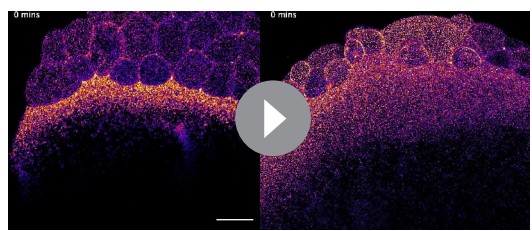

**Video 9.** Heterogenous distribution of F-actin in MZ*rab25b* EVL cells during late epiboly. Left Panel: WT embryo expressing Gfp-Utrophin (Fire-LUT) imaged beginning at 7hpf; Enrichment of cortical F-actin during marginal EVL cell rearrangements. Right panel: MZ*rab25b* embryo expressing Gfp-Utrophin (Fire-LUT) imaged beginning at 7hpf. Cells with reduced and large apices shorten their EVL-YC contact; Scale bar 20 μm.

https://elifesciences.org/articles/66060#video9

longer in mutant embryos compared to wild type, suggesting a more viscous response following laser ablation (*Figure 6F'*).

Overall, this suggests that forces are more balanced in the EVL and that viscoelastic responses are defective in Rab25 mutant embryos. Anisotropic tensions and contraction drive epithelial wound closure in *Drosophila* and epithelial monolayers in culture (*Zulueta-Coarasa and Fernandez-Gonzalez, 2018*). Thus, balanced forces would be predicted to disrupt force patterns in the EVL that drive oriented cell shape changes required for tissue elongation during epiboly progression. The overall reduction in actomyosin density and contractile networks likely contributes to the relaxation of tensions within the EVL of mutant embryos. Importantly, force is propagated across tissues through the actomyosin cytoskeleton. Accordingly, reduced actomyosin and tension would also be predicted to impact force transmission across the EVL during epiboly, disrupting the rate of tissue spreading. Lastly, inhibition of cell divisions in amniotes has been shown to increase viscosity, lending support to our data suggesting an association between cytokinesis failures and increased tissue viscosity (*Saadaoui et al., 2020*). Increased viscosity would slow the rate of EVL deformation in response to yolk cell pulling forces, given that viscous responses are slower than elastic deformations (*Petridou and Heisenberg, 2019*).

## MZ*rab25a* and MZ*rab25b* mutants exhibit endocytic trafficking defects

Rab25 is reported to direct recycling endosome pathways (*Casanova et al., 1999*). While our previous analysis of fluorescent Rab25 dynamics supports Rab25 trafficking in the EVL, we wanted to determine Rab25's position in the endocytic-recycling pathway. To do this, Venus-Rab25a or eGfp-Rab25b were co-expressed with mCherry-Rab11a, a marker of recycling endosomes (*Mavor et al., 2016*). Spatiotemporally, mCherry-Rab11a dynamics largely overlapped with Venus-Rab25a and eGfp-Rab25b over the duration of epiboly, with Rab11 and Rab25 constructs colocalizing within the cytosol and at the plasma membrane (*Figure 7A*, *Figure 7—figure supplement 1A*). During cell division, mCherry-Rab11a became enriched at centrosomes in parallel with fluorescently tagged Rab25 (*Video 11*, *Figure 7—figure supplement 1A*, black arrows). Given Rab11's established association with recycling endosomes (REs) (*van Ijzendoorn, 2006*), localization of Rab25 with Rab11 compartments during cell division and epiboly implies Rab25 localizes to REs in the zebrafish.

We next postulated that a vesicular trafficking defect may underlie the mutant phenotypes. To analyze this, Lyn-eGfp expressing WT and MZ*rab25a* and MZ*rab25b* embryos were incubated in pHRodo dextran. The fluorescently tagged peptide Lyn-eGfp labels intracellular membrane compartments during trafficking (*Sonal et al., 2014*). pHRodo dextran can only be internalized by EVL cells through apical endocytosis (*Figure 7E*), labeling endosomes positioned in endocytic pathways. In WT cells, we detected a small number of Lyn-eGfp cytoplasmic vesicular bodies with faint pHRodo dextran signal (*Figure 7B*). In contrast, both Lyn-eGfp- and pHRodo-positive vesicles accumulated in MZ*rab25a* and MZ*rab25b* embryos, consistent with defects in apical-endocytic membrane trafficking (*Figure 7B,F*). We also observed reduced junctional Lyn-eGfp in mutants compared to wild type, perhaps representing reduced fusion of vesicles with the plasma membrane, which we speculate could reflect recycling defects. Notably, the greatest accumulation of intracellular vesicles was observed in cells with large apices. Vesicles also accumulated in cells with normal morphologies and small apices, although to a lesser extent. These findings led us to hypothesize that the MZ*rab25a* and MZ*rab25b* phenotypes could be associated with trafficking defects and that these defects at least partially reflect the loss of Rab25 in mutant embryos as opposed to a secondary effect from changes in EVL cell morphology.

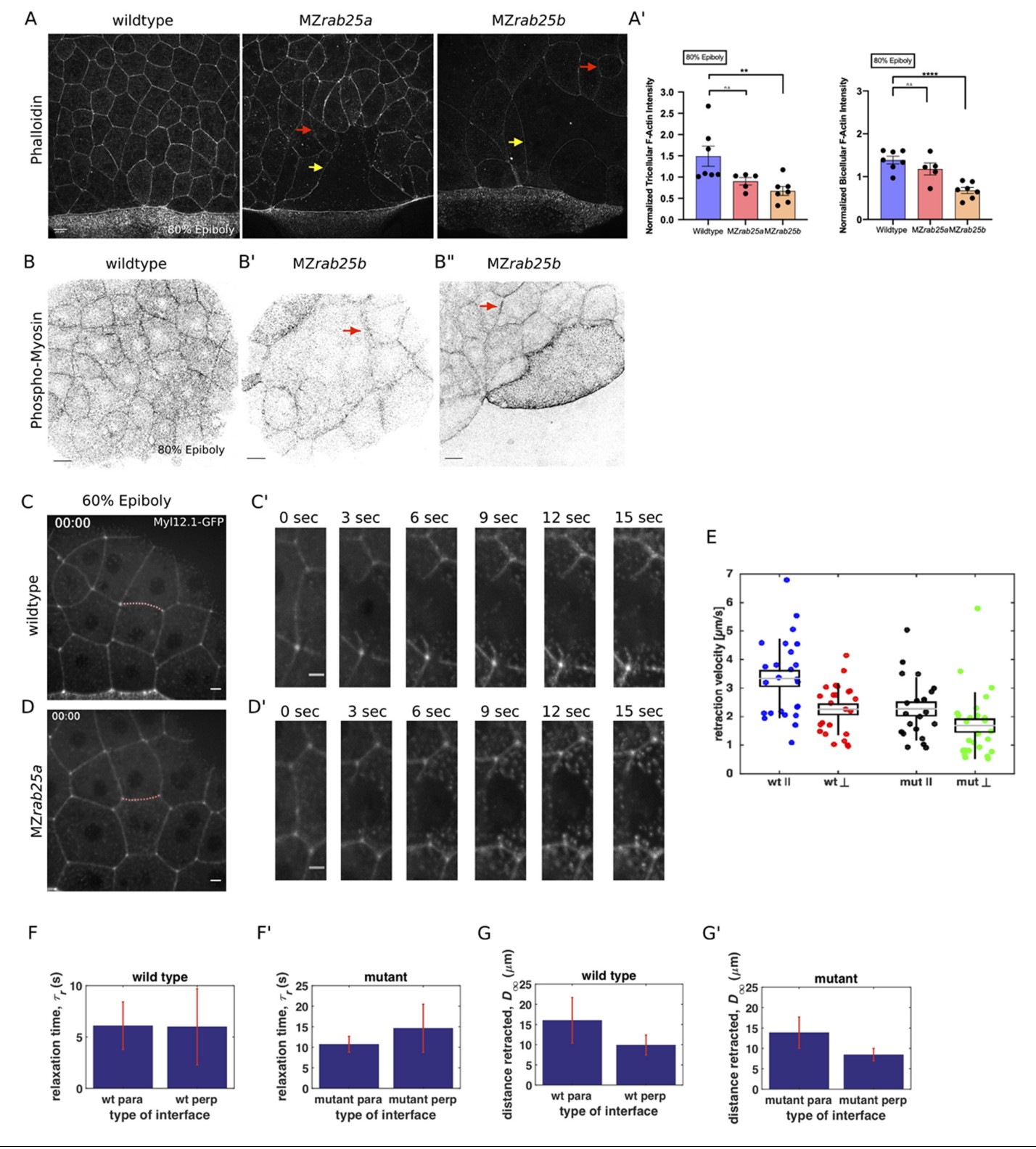

**Figure 6.** Reduced cortical actin and pMyosin associated with less tension and more viscous tissue responses in mutant embryos. (**A**) Lateral views with animal pole to the top of z-confocal projections of phalloidin stained WT, MZ*rab25a* and MZ*rab25b* embryos stage-matched at 80% epiboly; red arrows show reduced cortical actin in normal sized cells; yellow arrows show reduced actin in large cells; Scale bar 20 μm. (**A'**) Quantification of normalized tricellular and bicellular F-actin intensity at 80% epiboly. WT (n = 90,N = 9), MZ*rab25a* (n = 90,N = 9) and MZ*rab25b* (n = 90,N = 9). Means: SEM; Mann-

*Figure 6 continued on next page*

*Figure 6 continued*

Whitney, **,p<0.001. (B) Confocal z-confocal projections of WT and MZ*rab25b* embryos at 80% epiboly antibody stained for pMyosin. Red arrows denote uneven distribution of pMyosin along individual MZ*rab25b* cellular junctions. Scale bar 20 µm. (C–D') Confocal z-confocal projections of WT or MZ*rab25a* Tg(Myl1.1-Gfp) at 60% epiboly; lateral positioned embryo focused on EVL margin; red line marks the ablated junction. Scale bar 5 µm. (E–G') Initial recoil velocity, relaxation time and distance retracted in WT and MZ*rab25a* embryos following junction laser cutting (see Materials and methods). WT and MZ*rab25a* perpendicular and parallel cuts (n = 26,23);(n = 21,25).
The online version of this article includes the following figure supplement(s) for figure 6:

**Figure supplement 1.** Actomyosin intensity in MZ*rab25a* and MZ*rab25b* epiboly.

While accumulated dextran in mutants suggested internalized material was trapped in endosomes, we sought to further characterize these compartments in mutant embryos. We previously showed that Rab25 co-localized with Rab11, suggesting the excess vesicles may reflect recycling endosomes. To test this hypothesis, we used a Rab11 antibody to label recycling endosomes in WT and MZ*rab25b* embryos. Rab11b positive compartments were observed more frequently in EVL cells of MZ*rab25b* mutant embryos (40% of cells, n = 46/114, N = 8) compared to WT (6% of cells, n = 12/206, N = 6) (*Figure 7C,D*, yellow arrows). The size of Rab11-positive endosomes was also significantly increased in MZ*rab25b* embryos (*Figure 7—figure supplement 2*). Interestingly, in other systems, enlarged Rab11 recycling endosomes can be the result of reduced trafficking from REs to the cell surface (*Langevin et al., 2005*; *Mavor et al., 2016*). We also observed increased pHRodo dextran fluorescence intensity in some of the compartments in MZ*rab25b* mutant embryos, indicating they were acidic in nature (*Figure 7B,F*). This suggested that a subset of the accumulated vesicles may represent late endosomes or lysosomes, which could be the downstream result of an overall trafficking defect in mutant embryos. While further investigation is required, these data suggested that the increased number and size of vesicles in mutant embryos could partially be explained by an increase in recycling endosomes, which we postulate could result from a recycling defect.

## Discussion

Rab GTPases comprise the largest family of small GTPases and are an important class of membrane trafficking regulators (reviewed in *Nassari et al., 2020*). Recently, several Rabs have been shown to play roles in morphogenesis and embryonic development (*Jewett et al., 2017*; *Ossipova et al., 2014*; *Ossipova et al., 2015*; *Rathbun et al., 2020*). Rab25 is reported to be epithelial specific in mammals and is a member of the Rab11 subfamily (*Casanova et al., 1999*; *Goldenring et al., 1993*), which has been implicated in apical recycling, cytokinesis, junction reinforcement, and turnover (*Iyer et al., 2019*; *Langevin et al., 2005*; *Rathbun et al., 2020*; *Woichansky et al., 2016*).

Rab25 mutant mice are viable but exhibit defects in skin homeostasis suggesting role in epidermal formation (*Nam et al., 2010*; *Jeong et al., 2019*). The function of Rab25 during development is largely unexplored and here we present the first characterization of two *rab25* genes in the zebrafish embryo.

Zebrafish *rab25a* and *rab25b* are expressed in the surface epithelial layer, the EVL, during gastrulation. MZ*rab25a* and MZ*rab25b* mutant embryos exhibit epiboly delays and EVL cellular morphology defects. As described further below, we propose that the heterogeneous EVL cell sizes in mutant embryos, resulting from cytokinesis failures, impairs epiboly in two ways: by hindering local cell rearrangements and altering the viscoelastic properties of cells.

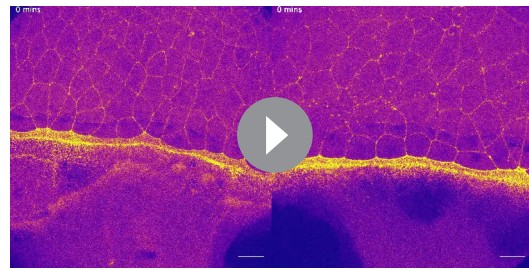

**Video 10.** Myosin-Gfp distribution in MZ*rab25b* embryo during epiboly progression. Left Panel: WT Tg (Myl1.1-Gfp) (Fire-LUT) embryo imaged starting at 7hpf. Right Panel: MZ*rab25b* Tg(Myl1.1-Gfp) (Fire-LUT) embryo imaged starting at 7hpf; Myosin-Gfp becomes heterogeneously distributed in EVL marginal regions Scale bar 20 µm.
https://elifesciences.org/articles/66060#video10

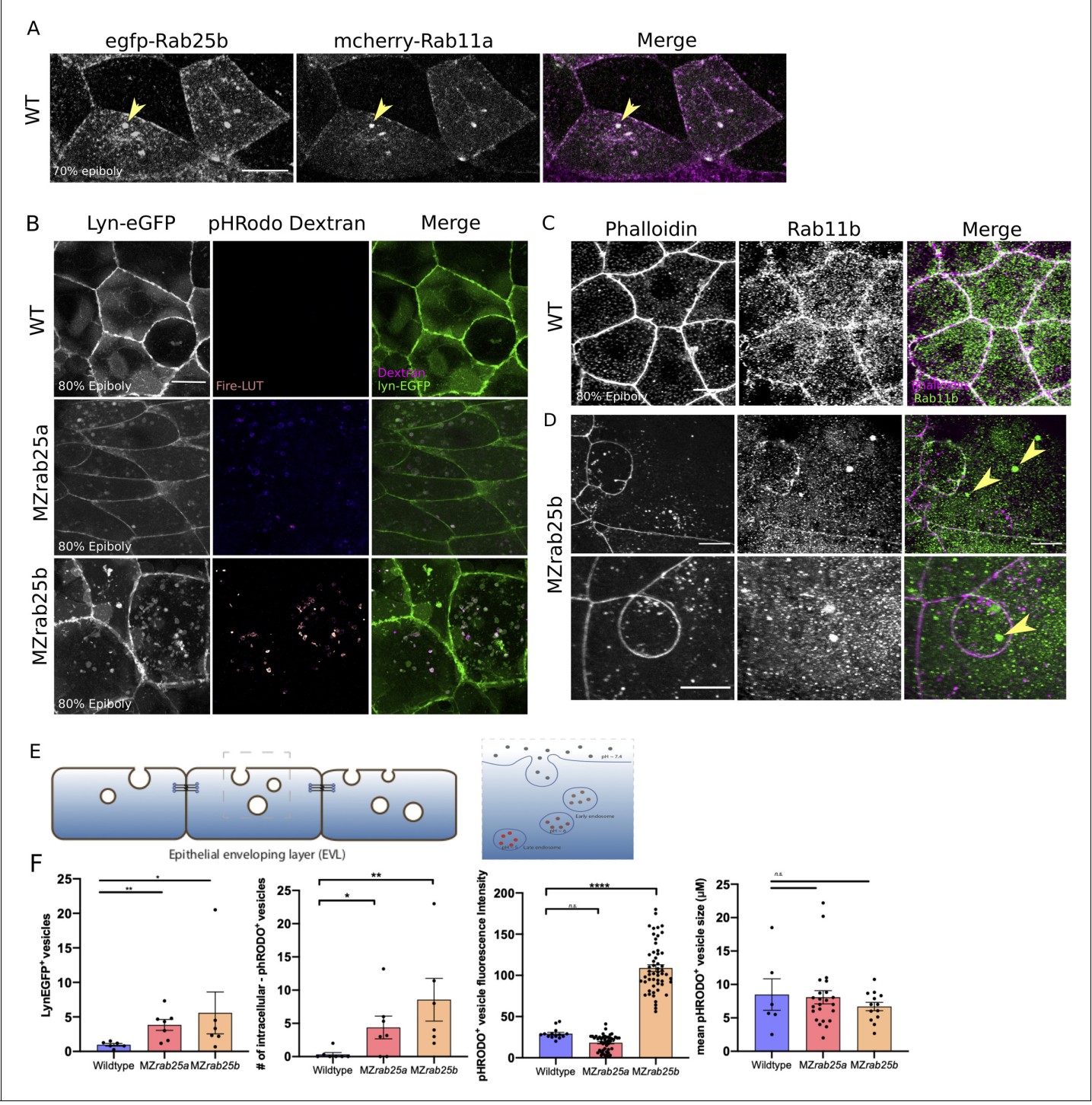

**Figure 7.** Co-localization of tagged-Rab25 with mCherry-Rab11a and MZ*rab25a* and MZ*rab25b* embryos exhibit trafficking defects. (**A**) z-projection stills of a wild-type embryo at 80% epiboly co-expressing mCherry-Rab11a (green) and eGfp-Rab25b (magenta); arrowhead denotes overlap. (**B**) Live WT, MZ*rab25a* and MZ*rab25b* embryos expressing Lyn-eGfp (green) and containing cytoplasmic pHRodo dextran puncta (magenta) following incubation. Scale bar 20 µm. (**C,D**) Rhodamine phalloidin stained (magenta) and Rab11b antibody (green) stained WT and MZ*rab25b* embryos at 80% epiboly; arrowheads denote large Rab11b endosomes. Scale bar 20 µm (**E**) Schematic of pHRodo dextran apical endocytosis (**F**) Mean number of Lyn-eGfp- or pHRodo-positive vesicles/cell. WT, MZ*rab25a* and MZ*rab25b* embryos (N = 7,7,6). Fluorescence intensity measured over a 1 µm line in pHRodo-positive vesicles; WT (n = 15); MZ*rab25a* (n = 46); MZ*rab25b* (n = 54). Mean surface area of pHRodo vesicles; WT(n = 6); MZ*rab25a* and MZ*rab25b* (n = 23,13). Means: SEM; significance using Mann-Whitney test. Scale bars, (**A–C**) 20 µm.

The online version of this article includes the following figure supplement(s) for figure 7:

*Figure 7 continued on next page*

*Figure 7 continued*

**Figure supplement 1.** Co-localization of N-terminally fluorescently tagged Rab11a, Rab25a, and Rab25b constructs and Rab11b endosome quantification.

**Figure supplement 2.** Quantifications of Rab11b-positive endosome diameter in WT (n = 18, N = 6) and MZ*rab25b* (n = 96, N = 8) embryos.

## Role for Rab25 in cytokinetic abscission

The overall subcellular distribution of Rab25 constructs is consistent with a function in trafficking (*Hehnly and Doxsey, 2014*; *van Ijzendoorn, 2006*; *Langevin et al., 2005*). Rab25 constructs were highly motile in the cytoplasm as puncta and transited toward the plasma membrane, suggestive of membrane and/or cargo delivery. During mitosis, Rab25 localized near centrosomes, where recycling endosomes are positioned to direct membrane to the intercellular bridge during cytokinesis (*Frémont and Echard, 2018*). Lastly, Rab25 co-localized with the known recycling protein Rab11.

MZ*rab25b* embryos exhibited a range of cytokinesis defects during epiboly which led to the formation of bi-and multinucleate cells. Unstable cytokinetic bridges were associated with failed formation of cytokinetic midbodies while persistent apical cytokinetic bridges experienced either failed or delayed abscission. Binucleate cells remained mitotically active generating large, multinucleate cells with reduced cortical actin which continued to exhibit abscission defects leading to an array of interconnected cells across the EVL. These observations point to a previously uncharacterized role for Rab25 in cell division.

Cell division and cytokinesis are multicellular processes (*Herszterg et al., 2013*), and morphogenic stress in the form of neighboring cell divisions or basal cell extrusion events appeared to tear open persisting cytokinetic bridges in mutant embryos. In cell culture, when abscission is delayed through siRNA knockdown of Rab35, mitosis similarly tears open cytokinetic bridges (*Frémont et al., 2017*). Our observations of a few cells failing to initially form bridges and a large proportion of cells experiencing delayed abscission in MZ*rab25b* mutants is highly similar to the phenotypes of Rab11 and Rab35 knockdown experiments in vitro (*Frémont et al., 2017*; *Kouranti et al., 2006*; *Mierzwa and Gerlich, 2014*). Furthermore, optogenetic clustering of Rab11 during Kupffer's vesicle morphogenesis in zebrafish resulted in abscission failure (*Rathbun et al., 2020*). Therefore, our data implicates Rab25 in having a direct role in cytokinesis.

Current models of membrane dynamics during intercellular bridge cleavage suggest coordination amongst many Rab recycling pathways (*Frémont and Echard, 2018*). Rab11 and Rab35 are thought to act in a redundant manner in regulating bridge scission in vitro (*Frémont et al., 2017*), although whether this is the case in animal models has yet to be determined. Our evidence suggests Rab25a and Rab25b have similar functions during epiboly. We propose this as *rab25b* transcripts were upregulated in MZ*rab25a* embryos which was associated with less severe EVL cytokinesis defects.

Rab11 vesicle trafficking has been shown to be important for cytokinesis during zebrafish KV development (*Rathbun et al., 2020*). As the KV is an EVL derived organ (*Oteíza et al., 2008*), it is likely Rab11 has an abscission function during epiboly. Previous Rab11 dominant negative or morpholino approaches have not produced phenotypes similar to MZ*rab25b* mutant embryos, but this can likely be attributed to maternally expressed Rab11 family members rescuing the depletion of zygotic Rab11 in these experiments (*Clark et al., 2011*). Rab11 and Rab25 may be acting stepwise in vesicle delivery toward the cytokinetic midbody to regulate abscission or may regulate separate trafficking pathways altogether. A more attractive hypothesis is that Rab11 and Rab25 work as a complex to regulate abscission or have partially overlapping trafficking pathways, a proposal consistent with both constructs co-localization and parallel dynamics during epiboly. Coregulation of Rab proteins in

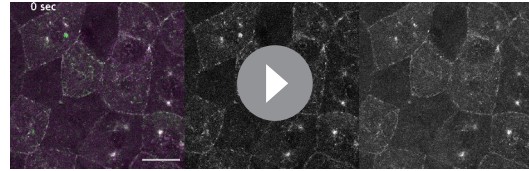

**Video 11.** mCherry-Rab11a spatiotemporally overlaps with Venus-Rab25a in the cytosol, plasma membrane, and at centrosomes. Left Panel: Confocal time-lapse of a WT embryo expressing mCherry-Rab11a (magenta) and Venus-Rab25a (green) beginning at 30% epiboly. Middle Panel: Venus-Rab25a. Right Panel: mCherry-Rab11a. Scale Bar 20 μm.
https://elifesciences.org/articles/66060#video11

trafficking pathways is observed in in vitro models (reviewed in *Stenmark, 2009*).

An open question is why Rab25a or Rab11 cannot compensate for loss of Rab25b in MZ*rab25b* mutant embryos. A possible explanation may be that more Rab25b protein is maternally deposited compared to Rab11a or Rab25a, thus, disruption of Rab25b without transcriptional adaptation of *rab25a* or *rab11a* may not be able to rescue the abscission phenotypes. Given the lack of Rab25b antibody this is difficult to determine. Membrane trafficking becomes increasingly important in morphogenetically active tissues, as a higher degree of plasma membrane remodeling occurs (reviewed in *Pinheiro and Bellaïche, 2018*). This is exemplified in the related teleost fish Fundulus, where membrane turnover in the EVL was shown to steadily increase as epiboly progressed (*Fink and Cooper, 1996*), suggesting that membrane recycling may similarly increase. Thus, having high amounts of Rab trafficking molecules during epiboly is most likely needed for normal cytokinesis.

## Cytokinesis failures disrupt EVL morphogenesis

Cell division appears to have two major roles in epithelial morphogenesis. Cell-division-mediated intercalations (CMI) power epithelial rearrangements in chick, quail, and *Xenopus* embryos during gastrulation (*Firmino et al., 2016*; *Higashi et al., 2016*; *Saadaoui et al., 2020*). In general, cell rearrangements have well-defined roles in tissue morphogenesis and embryonic development (reviewed in *Zallen and Blankenship, 2008*). Thus, failed CMI disrupts rearrangements critical for tissue development, as shown in amniote gastrulation and mouse limb bud development (*Firmino et al., 2016*; *Lau et al., 2015*; *Saadaoui et al., 2020*; *Wyngaarden et al., 2010*). A second role of cell divisions is the regulation of tissue fluidity and viscoelasticity (reviewed in *Petridou and Heisenberg, 2019*). Recent experiments using laser ablation and micropipette assays on amniote epiblast cells and zebrafish deep cells have shown that inhibiting cell division increases tissue viscosity, effectively slowing tissue shape changes (*Petridou et al., 2019*; *Saadaoui et al., 2020*). Our data in MZ*rab25a* and MZ*rab25b* embryos suggests that failed cytokinesis disrupted cell intercalation events during epithelial epiboly and increased tissue viscosity, with both likely contributing to slow EVL spreading.

Marginal epithelial cell rearrangements are required for normal epiboly movements of surface epithelia in zebrafish, Fundulus, and Tribolium embryos (*Jain et al., 2019*; *Keller and Trinkaus, 1987*; *Köppen et al., 2006*). Similar to T1 transitions and rosette resolution described in the *Drosophila* germband (*Blankenship et al., 2006*), marginal EVL cell intercalations bring into contact initially non-neighboring cells. Marginal intercalation leads to EVL circumference narrowing needed to close the blastopore (*Köppen et al., 2006*). The EVL cell shape heterogeneity in mutant embryos led to an overall slowing of marginal cell intercalation events, which likely impaired EVL circumference narrowing and the rate of epiboly in MZ*rab25a* and MZ*rab25b* embryos. While we investigated cell rearrangement events predominantly in EVL marginal regions, our time-lapses of wild-type embryos revealed EVL cell neighbor exchanges throughout most regions of the epithelium, which to date has not been reported.

## Rab proteins and cytoskeletal dynamics

Tension anisotropy promotes tissue shape changes during morphogenesis. For example, heterogenous contractile networks are required for timely wound healing and apical cell extrusion (*Ap et al., 2020*; *Zulueta-Coarasa and Fernandez-Gonzalez, 2018*). Our laser cutting analysis revealed polarized tensions within the EVL, with cell junctions aligned parallel to the EVL-YC boundary having high recoil velocities compared to perpendicular junctions elongating during epiboly progression. Higher tensions along parallel contacts indicate cells may dynamically narrow along the circumferential embryonic axis during late-phase epiboly, consistent with the tissue narrowing and elongating. It may also mean some junctions are resisting external forces to maintain their initial length. In support of our notion that tension heterogeneity drives EVL morphogenesis, the overall relaxation of tensions in mutant embryos disrupted polarized force patterns which was associated with disorganized and stochastic cell behaviors that hindered collective cell movements and epiboly.

Similar to what has been described in several different *Drosophila* epithelia (*Lecuit et al., 2011*), we found that abnormal cell shape changes were associated with a global reduction of cortical actomyosin networks in MZ*rab25a* and MZ*rab25b* embryos. We observed that enlarged, multinucleated cells had the largest depletions of cortical actomyosin. This reduction in cortical actomyosin provides a likely explanation for the significant slowing of cell rearrangements exhibited by multinucleate cells

compared to neighboring cells with normal or reduced apical surface areas. Furthermore, this finding aligns with the observation that greater epiboly delays were associated with the increased presence of multinucleated cells. Reduced cortical actin also explains the tension defects shown via laser ablation, which we also propose results in perturbed force transmission across the EVL. Despite the relationship observed between cell size and cortical actin density, cytokinesis failures do not explain weak actomyosin intensity throughout the EVL in normal sized cells in both mutant backgrounds. This suggests that Rab25 may regulate the cortical actin network independently of its role in abscission.

Rab proteins have conserved roles in actomyosin network maintenance. During *Drosophila* mesectoderm invagination and germband extension, Rab35 localizes to actomyosin networks and is required for sustained cell-cell interface contraction (*Jewett et al., 2017*). Disruption of Rab35 impairs cell shape changes and overall tissue morphogenesis in *Drosophila* (*Jewett et al., 2017*). In *Xenopus* bottle cell ingression and neuroepithelial folding, Rab11 similarly promotes actomyosin contractility to drive both developmental programs (*Ossipova et al., 2014*; *Ossipova et al., 2015*). Thus, Rab25 having a role in actomyosin regulation during epiboly is consistent with Rab protein functions in other systems.

The mechanism for Rab25 cortical actomyosin maintenance is currently unclear and requires further study. Rab proteins promote cytoskeletal changes at subcellular sites by delivering actin remodeling proteins. For example, during abscission, Rab11 delivers p50RhoGAP to depolymerize intercellular bridge actin networks (*Schiel and Prekeris, 2013*). Additionally, a Rab11 effector, Nuf, is involved in actin polymerization during *Drosophila* cellularization (*Cao et al., 2008*). Rab25 vesicles may also contain lipids involved in signaling pathways which reorganize actin networks. For example, Rab35 vesicles contain the PtdIns(4,5)P2 phosphatase OCRL which remodels F-actin during abscission (*Dambournet et al., 2011*).

Alternatively, junctional reinforcement at cell vertices may lead to downstream actomyosin polymerization and contractility. In *Drosophila,* recycling pathways have well defined roles in cell contact reinforcement, thus Rab25 may regulate the actomyosin cytoskeleton during epiboly in a similar fashion. Studies in *Xenopus* have shown that Rab11 recruitment to vertices is modulated by tension (*Ossipova et al., 2014*). In a positive feedback loop, Rab11 then further promotes actomyosin contractility at these sites to drive apical constriction (*Ossipova et al., 2014*). Given that tagged-Rab25 localized to vertices, which are known to be under high tension during cell rearrangements (*Bosveld and Bellaïche, 2020*), Rab25 may similarly redistribute in response to tension, with the downstream effect of promoting contractility and actin polymerization. Future endeavours will help determine this mode of Rab25 function during epiboly.

We propose a two branched model by which Rab25 functions during epithelial epiboly in the zebrafish gastrula. Rab25 trafficking coordinates cytokinetic abscission to ensure successful EVL cell divisions, which is critical for proper cell size, shape and number. In parallel, Rab25 regulates cortical actomyosin network density in the EVL. Increases in cell size from failed cytokinesis led to sparse cortical actin density, thus both Rab25 pathways impact cortical cytoskeleton density in the EVL. Maintenance of the actin cytoskeleton is required for orderly cell rearrangements, polarized force patterns, and viscoelastic responses to ensure normal epiboly progression.

## Materials and methods

### Zebrafish handling

Animals were maintained under standard conditions in accordance with the policies and procedures of the University of Toronto animal care committee. Fish stocks were housed at 28–29°C in an Aquaneering Zebrafish Housing System with a pH from 7.2 to 7.8 and conductivity between 500 and 700 µS. Adults used were on average 1 year old with no prior manipulations or apparent health issues. Embryos were collected from natural spawnings using a tea strainer and rinsed and stored in facility water or E3 medium at 28.5–30°C. Embryos were staged as described by *Kimmel et al., 1995*.

## Quantification and statistical analysis

Quantifications represent data collected from cells visualized within a given area illuminated by confocal microscopy and not estimates of embryo wide measurements.

### Epiboly Progression

To assess the extent of epiboly progression, the leading edge of the blastoderm was labeled using either phalloidin or by in situ hybridization against ta (*ntla*). Embryos were photographed and ImageJ was used to measure the distance from the animal pole to EVL leading edge and compared to the overall length of the embryo along the animal-vegetal axis (AV) (SF 1D).

### Cell contact analysis

The number of bicellular contacts between an EVL cell and its neighbors EVL cells or the YC was counted.

### Apical surface area, circularity, and heat maps

Surface area, circularity and heat maps were calculated and generated using SIESTA and custom scripts written in MATLAB (*Zuluerta-Coarasa et al., 2017*).

$$\text{Circularity} = p^2/4\pi a,$$

where p is the cell apical surface perimeter and $\alpha$ is cell apical surface area. Circularity is one for circles and larger than one for non-circular shapes.

### Phalloidin fluorescence intensity

Phalloidin levels were measured in fixed embryos in EVL marginal regions at the stages described using a 2 µM line at bicellular and tricellular contacts. For each experiment, all measurements were then corrected and normalized to ZO-1 levels in the same regions using the measure tool in ImageJ.

### Myosin intensity

Myosin levels were measured across a 2 µM line in the middle of bicellular junctions in live embryos expressing Myl12.1-eGFP pre-ablation. For each experiment, all measurements were then corrected and normalized to myosin intensity at tricellular contacts along the same junction using the measure tool in ImageJ.

### pHRodo dextran and Lyn-eGFP vesicle counts and fluorescence intensity

In each experiment, vesicles > 0.75 µM were counted in images from live confocal z-projections of EVL cells. To measure the fluorescence intensity of pHRodo positive vesicles, intensity across a line spanning the diameter of the vesicles was measured. Analysis was done using the measure tool in ImageJ.

### Cytokinetic bridge length and duration

Bridge length and duration in wild-type embryos was quantified in marginal and animal regions in live AB embryos expressing Dcx-Gfp from plasmid injection; Tg:(XlEef1a:eGFP-tubα8l) or; Tg:(XlEef1a1:Dclk2DeltaK-GFP). All methods produced similar bridge lengths and durations. Mutants embryos expressing Dcx-Gfp from plasmid injection were used for bridge analysis. Bridge length for all genotypes were measured following intercellular bridge formation. Bridge duration analysis began following cleavage, when microtubules were first organized into bridges. Duration analysis ended when microtubules were no longer visible, indicating abscission. Duration analysis also ended in mutants if bridges were present at the end of time-lapses or when bridges regressed or were torn open. Initial bridge length following formation was measured using the measure tool on ImageJ.

### Spindle orientation

The angle of EVL mitotic spindles was compared to the long axis of the cell in live wild-type and mutant embryos in marginal and animal regions using the measure tool in ImageJ. Microtubules in wild-type embryos were labeled by Dcx-Gfp from plasmid injection; Tg:(XlEef1a:eGfp-tubα8l) or; Tg:

(XIEef1a1:dclk2DeltaK-Gfp). In mutant embryos, microtubules were labeled with Dcx-Gfp from plasmid injection. Embryos also expressed membrane-Rfp and H2A-Rfp.

## EVL-YC contact shortening
The length of EVL-yolk cell contact was measured each frame over the duration of time-lapses using the measure tool in ImageJ.

## Multicellular EVL-YC vertex resolution
Following formation of multicellular vertices along the EVL-YC boundary, resolution of rosettes was determined when a new cell contact was observed to form between cells adjacent to intercalating cells.

## Rab11b recycling endosome number and size
To assess the number of Rab11b-positive compartments, Rab11b-positive endosomes with diameters > 1 µM were counted in individual EVL cells of WT and MZ*rab25b* embryos stained with Rab11b antibodies and phalloidin. The diameter of Rab11 compartments was measured to compare the sizes of Rab11 positive endosomes between genotypes and was performed using the measure tool in ImageJ.

## Statistics
Statistical Analysis was performed using PRISM software. Sample means p values were compared to infer significance, with the type of test indicated in figure captions.

## CRISPR/Cas9 mutant generation
To determine Cas9 target sites against rab25a and rab25b (ZFIN ID: ZDB-Gene-041212–69; ZDB-Gene-050706–113), the CHOPCHOP tool was used (https://chopchop.cbu.uib.no). The following target sites were chosen for *rab25a*: Exon 2 (Rab GTPase Domain), 5'-AGTGGTTTTAATTGGAGAATCAGG-3'; for *rab25b*: Exon 2 (Rab GTPAse Domain), 5'-CTGGATTGGAGCGGTACCGC −3'. To generate gRNA's, sgDNA templates were generated using PCR without cloning:

> Oligo 1
> T7 Promoter PAM *rab25a* Target Site Overlap with oligo 2
> 5'TTAATACGACTCACTATA|GGA|GTGGTTTTAATTGGAGAATCAGGGG|GTTTTAGAGCTAGAAATAG-3'
> T7 Promoter PAM *rab25b* Target Site Overlap with oligo 2
> 5'-TTAATACGACTCACTATA|GGC|GTGCGCTCCGTTCCTACAGAGG|GTTTTAGAGCTAGAAATAG
> Oligo 2-
> 5'-AAA AGC ACC GACTCG GTG CCA CTT TTT CAA GTT GAT AAC GGA CTA GCCTTATTT TAA CTT GCT ATT TCT AGCTCT AAA AC-3'

The Ambion MegaScript T7 kit was used to transcribe sgRNA in vitro. gRNA (50 pg) was coinjected with *cas9* mRNA (300 pg)(pT3TS-nCas9 [Xba1 digest] Addgene plasmid: #46757 *Jao et al., 2013*; transcribed using mMESSAGE mMachine T3 kit Life technologies [AM1348]). The Zebrafish Genetics and Disease Models Core Facility at the Hospital for Sick Children generated founder Rab25a CRISPR zebrafish using the guide RNA that we designed.

To determine indel frequencies, genomic DNA from 24hpf injected embryos was extracted using Lysis Buffer and Phenol Chloroform purification with the following primer pairs used for amplification target sites:

> rab25a
> Forward-5'- TATTTATTCACCAAGCGGTTG-3'
> Reverse-5'- GAGTGGTTCTGGGTGTGAGTC-3'
> rab25b
> Forward- 5'-TGTTTGCAGTGGTTCTTATTGGAG-3'
> Reverse- 5'-ATTACGTTCGCTTGCAGAATTT-3'

rab25a and rab25b PCR products were digested with Hinf1 and Kpn1, respectively. PCR products which could not be cut were then sequenced using the respective forward primers above. This led to the identification of two *rab25a* mutations from two different founder fish and one *rab25b* mutation:

*rab25a$^{2.3}$*–13 bp deletion 5'-TTGCTTTCAGTGGTTTTAATTGGAGAA————————————
AG-3' *rab25a$^4$*- 24 bp insertion / 3 bp deletion
5'TTGCTTTCAGTGGTTTTAATTGGAGAATTGGAAAGTTGGAAAGCGCAACTTTGGG
TTGGGTTGGAAAG-3' *rab25b*- 18 bp deletion 5'-TG——————————CCATCACCTCTGCG
TGAGTTTG-3'

We had difficulty rescuing the mutant phenotypes by RNA injection, likely due to the required maternal contribution of *rab25a* and *rab25b*. We also observed that *rab25a* mutant alleles produced more severe phenotypes over successive generations of in-crossing.

## Cloning

AB, MZ*rab25a*, or MZ*rab25b* embryos at 24hpf were dechorionated and RNA was extracted from 60 to 80 embryos using Trizol (Invitrogen). cDNA was generated using Postscript II first strand cDNA synthesis kit (NEB). Coding sequences were PCR amplified from cDNA or vectors with Q5 high-fidelity Taq Polymerase (NEB) using the following primers:

*rab25a:* pGEM
Forward: 5'ATGGGGACAGATTTAGCCTACAAC-3'
Reverse: 5'CGAAGCTGCTGCAAAAACTCCTGA-3'

PCR products were gel purified and A-tailed by incubation with dATP and Taq Polymerase (NEB) for 30 min and ligated into pGEM-T-Easy (Promega).

rab25a: pCS2+ *rab25a* was PCR amplified from pGEM with PCR primers containing Cla1/Stu1 restriction sequences for forward and reverse primers, respectively. PCR fragments were digested with either Cla/Stu1, gel extracted and ligated into pCS2+. Constructs were confirmed by sequencing.

rab25a: venus-pCS2+
Forward: 5'-CTCGAGGGCGCCACCATGGGGACAGATTTAGCCTACAAC-3'
Reverse: 5'- CTCGAGCGAAGCTGCTGCAAAAACTCCTGA-3'

N-terminal venus fusion protein was generated by digesting vector and insert with Xho1 and ligated. Sequencing was used to validate ligation of venus and *rab25a*. rab25b:

Forward:
5'-GGGGACAAGTTTGTACAAAAAAGCAGGCTTCATGGGGTCTGATGAGGCCTA-3'
Reverse:
5'-GGGGACCACTTTGTACAAGAAAGCTGGGTGTCACAAGTTTTTACAGCAGG-3'

## Rab25b fusion proteins

To generate Rab25b fusion proteins, forward and reverse primers containing *attB1* and *attB2* BP Clonase recognition arm sequences were used to PCR amplify *rab25b*. *attb*-tagged *rab25b* PCR Products were gel extracted and recombined into a pDONR221 entry vector using BP Clonase (Invitrogen 11789013). Clones were identified via kanamycin selection and validated by sequencing.

*attB1rab25b* Forward Primer:
5'GGGGACAAGTTTGTACAAAAAAGCAGGCTTCATGGGGTCTGATGAGGCCTA-3'
*attB2rab25b* Reverse Primer:
5'GGGGACCACTTTGTACAAGAAAGCTGGGTGTCACAAGTTTTTACAGCAGG-3'

pDONR221-*rab25b* vectors were recombined into a pCS2+ SP6 promoter destination vector by Gateway LR Clonase reaction (Invitrogen 11791020). Destination vectors contained a 5' SP6 promoter sequence, followed by either eGfp or mCherry with Rab25 integrated in the 5' to 3' direction downstream, resulting in a N-terminally tagged Rab25b fusion protein. Sequencing was used to confirm integration and reading frame.

## FP2 constructs

The constructs pFP2-DeAct-SpvB, pFP2-DeAct-GS1 and pFP2-DN-RhoA were generated to block actin polymerization/contraction in the yolk cell. pCMV-DeAct-SpvB (Addgene plasmid 89446) and pCMV-DeAct-GS1 (Addgene plasmid 89445) were gifts from Bradley Zuchero (*Harterink et al., 2017*) and pFP2 was kindly provided by Arne Lekven (*Narayanan and Lekven, 2012*). Both DeActs were PCR amplified and cloned into pCS2+. SpvB was amplified using primers containing Cla1/Stu1 restriction sequences for forward and reverse primers, respectively. PCR fragments were digested with either Cla/Stu1, and ligated into pCS2+. GS1 was amplified using primers containing BamHI/ StuI restriction sequences for forward and reverse primers, respectively. PCR fragments were digested with either BamHI/StuI, and ligated into pCS2+. SpvB, GS1 and DN-Rho were cloned from pCS2+ into pFP2 using the Gibson assembly method (*Gibson et al., 2009*).

Primers used to amplify DeActs were:

SpvB Forward primer:
5'- ACGATCGATGCCACCATGGGAGGTAATTCATCTCG-3'
SpvB Reverse primer:
5'-GACAGGCCTTCATGAGTTGAGTACCCTCA-3'.
GS1 Forward primer:
5'-ACGGGATCCGCCACCATGGTGGTGGAACACCCCGA-3'
GS1 Reverse primer:
5'-CCGAGGCCTTCAGAATCCTGATGCCACAC-3'.

Primers used to clone DeActs and DN-RhoA from pCS2+ into pFP2 were:

Forward primer:
5'-GGTCACTCACGCAACAATACAAGCTACTTGTTCTTTTTG-3'
Reverse primer:
5'-CATGTCTGGATCATCATTACGTAATACGACTCACTATAG-3'

# Transgenic lines

## Rab25b Transgenic rescue line

To generate a transgenic rescue construct for Rab25b, forward and reverse primers containing *attB1* and *attB2* BP Clonase recognition arm sequences were used to PCR amplify *rab25b. attb*-tagged *rab25b* PCR Products were gel extracted and recombined into a pDONR221 entry vector using BP Clonase (Invitrogen 11789013). Clones were identified via kanamycin selection and validated by sequencing.

*attB1rab25b* Forward Primer:
5'GGGGACAAGTTTGTACAAAAAAGCAGGCTTCATGGGGTCTGATGAGGCCTA-3'
*attB2rab25b* Reverse Primer:
5'GGGGACCACTTTGTACAAGAAAGCTGGGTGTCACAAGTTTTTACAGCAGG-3'

pDONR221-*rab25b* vectors were recombined into a Tol2 transgenic destination vector by Gateway LR Clonase reaction (Invitrogen 11791020). Destination transgenic vectors contained two Tol2 recognition sequences that flanked divergent ß-actin and *myl7* promoter sequences. Rab25b was integrated in the 5' to 3' orientation downstream of the ß-actin promoter sequence. Sequencing was used to confirm integration and reading frame. The *myl7* promoter sequences contained a downstream RFP expression cassette for screening purposes. To generate Tg(*actb1:rab25b,myl7:RFP*) fish, Tol2 mRNA (25pg) and the destination vector *Tol2-RFP-myl7*:ß-actin-*rab25b-Tol2* (50pg) were injected into 1-cell stage embryos. Embryos were screened at 48 hpf to confirm Myl7 heart restricted fluorescence and grown to adulthood.

## Rab25a Myosin Transgenic Line

Female Tg(*actb2:myl12.1-eGFP*) fish were crossed to *rab25a*[4] homozygous males. Heterozygous transgenic embryos were screened for fluorescence at 24hpf and grown to adulthood. Tg:(*actb2: myl12.1-eGFP*, *rab25a*[4] (+/-)) fish were in-crossed to generate Tg:(*actb2:myl12.1-eGFP*, *rab25a*[4](-/-) adult fish, which were screened for fluorescence at 24hpf and genotyped for the *rab25a* mutation by PCR/Hinf1 restriction digest.

## mRNA synthesis and microinjections

Unless specified otherwise, Not1-digested plasmids were used as templates for in vitro transcription using the SP6 mMessage mMachine Kit (Ambion). mRNA was purified using MEGAclear kit or Nuc-Away Spin Columns (Ambion). RNAs were injected into the yolk cell or blastoderm of one-cell stage embryos, as described (*Bruce et al., 2003*). Doses of injected RNA were: *h2a-rfp* (50 pg), *mrfp/gfp* (50 pg), *lyn-egfp* (50 pg), *gfp-Utrch* (100 pg), *mcherry-rab11a* (300 pg), *egfp-rab25b* (300 pg), *venus-rab25a* (300 pg), *mcherry-rab25b* (300 pg), *mcherry-mklp1* (150–200 pg). Plasmids were injected at doses ranging from 10 to 20 pg.

## Whole-mount immunohistochemistry

Antibody staining was performed as previously described (*Lepage et al., 2014*). Dilutions were as follows: rhodamine-phalloidin (1:200), anti-E-cadherin (Abcam, 1:1000), anti-RAB11B (Abcam 1:200), anti-ZO-1 (ThermoFisher Scientific, 1:500), anti-phospho-myosin-light chain 2 Ser 19 (Cell Signaling, 1:100). Embryos were mounted in either 80% glycerol or 0.05% low-melt agarose. Secondary antibodies used were goat-anti-mouse Alexa 488 (Invitrogen A11001, 1:500), goat anti-rabbit Alexa 488 (Invitrogen A11008, 1:500), goat anti-rabbit-Cy3 (Jackson ImmunoResearch AB-2338006,1:500). Sytox green (Invitrogen) was diluted to 0.5 mM in fixative. For embryos co-labelled with phalloidin and antibody, phalloidin was incubated at 1:200 dilution with primary antibody overnight at 4°C in block solution.

## Whole-mount in situ hybridization

Whole mount in situ hybridizations were performed as previously described (*Jowett and Lettice, 1994*). *keratin4, ta, rab25a, rab25b, gsc, sox17* probes were generated by linearizing pBS vectors and transcribing using T7 RNA polymerase (ThermoFisher Scientific). Probes were purified using NucAway Spin Columns (Ambion).

## pHRodo dextran assay

Embryos were injected into the yolk cell at the one-cell stage with *lyn-egfp* mRNA, dechorionated and bathed in pHRodo red dextran (Invitrogen, P10361) dissolved in E3 medium (100 µg/ml). WT and *MZrab25b* embryos were incubated in the same media, while *MZrab25a* embryos were in a separate dish. Treatments were performed in petri dishes and embryos were incubated from high stage until 90% epiboly at 30°C. *MZrab25b* embryos were identified by their phenotype and mounted laterally in 0.05% low-melt agarose on glass bottom dishes (Matek).

## Laser ablations

Tg:(actb1:myl12.1-eGfp) and MZ*rab25a* Tg:(actb1:myl12.1-eGfp) embryos at 60% epiboly were dechorionated and mounted laterally in 0.05% low-melt agarose on glass bottom dishes (Matek). Imaging was done at room temperature using a Revolution XD spinning disk confocal microscope (Andor Technology) with an iXon Ultra 897 camera (Andor Technology), a 40X (NA1.35; Olympus) oil-immersion lens and Metamorph software (Molecular Devices). Ablations were performed in marginal regions, 2–3 cell rows back from the EVL-yolk cell margin on junctions parallel and perpendicular to the margin. Junctions were cut using a pulsed Micropoint nitrogen laser (Andor technology) tuned to 365 nm. The tissue was imaged immediately before and after ablation in which 10 laser pulses were delivered. 16-bit z-stacks were acquired every 3 s at 3 µm stacks and projected for analysis.

## qPCR

Total RNA was purified from shield stage of zebrafish embryos using Trizol, and an additional DNase I (Turbo DNaseI from Thermo Fisher Scientific) step was used to remove genomic DNA. RNA was reverse-transcribed with random primers using the high-capacity cDNA synthesis kit (Thermo Fisher Scientific). Gene expression was monitored by quantitative real time PCR (qPCR) using primers that distinguish individual transcripts (Key Resources Table). The standard curve method was used to calculate expression levels, with zebrafish genomic DNA used to generate the standard curves. Levels of *lsm12b* RNA were used to normalize expression values; primer sequences are shown in the Key Resources Table. All samples were confirmed not to have DNA contamination by generating a

reverse transcriptase negative sample and confirming there was no *lsm12b* amplification. At least three biological replicates were analyzed for each experiment. Significant differences in gene expression between shield stage embryos were determined by t-test.

## Imaging

Imaging was performed on a Leica TCS SP8 confocal microscope unless otherwise specified.

## Acknowledgements

We are grateful to Heidi Hehnly for the Rab11 constructs. We thank Curtis Boswell for technical advice. For helpful discussions and comments on the manuscript, we thank Rudi Winklbauer, Tony Harris, Vince Tropepe, Ulli Tepass and members of the Bruce lab.

## Additional information

### Funding

| Funder | Grant reference number | Author |
| --- | --- | --- |
| Natural Sciences and Engineering Research Council of Canada | RGPIN-2018-04862 | Ashley EE Bruce |
| Canadian Institutes of Health Research | 156279 | Rodrigo Fernandez-Gonzalez |
| Natural Sciences and Engineering Research Council of Canada | RGPIN-2020-05972 | Jennifer A Mitchell |

The funders had no role in study design, data collection and interpretation, or the decision to submit the work for publication.

### Author contributions

Patrick Morley Willoughby, Conceptualization, Data curation, Formal analysis, Validation, Investigation, Visualization, Methodology, Writing - original draft, Project administration, Writing - review and editing; Molly Allen, Roman Korytnikov, Tianhui Chen, Yupeng Liu, Isis So, Neil Macpherson, Investigation, Writing - review and editing; Jessica Yu, Data curation, Formal analysis, Validation, Investigation, Visualization, Methodology, Writing - review and editing; Haoyu Wan, Resources, Methodology, Writing – reviewing and editing; Jennifer A Mitchell, Resources, Funding acquisition, Writing - review and editing; Rodrigo Fernandez-Gonzalez, Resources, Data curation, Software, Formal analysis, Supervision, Funding acquisition, Validation, Visualization, Writing - original draft, Writing - review and editing; Ashley EE Bruce, Conceptualization, Resources, Data curation, Supervision, Funding acquisition, Validation, Investigation, Methodology, Writing - original draft, Project administration, Writing - review and editing

### Author ORCIDs

Patrick Morley Willoughby https://orcid.org/0000-0002-9051-8916
Molly Allen http://orcid.org/0000-0002-2419-0000
Tianhui Chen http://orcid.org/0000-0003-2236-2224
Isis So http://orcid.org/0000-0001-6232-2644
Jennifer A Mitchell https://orcid.org/0000-0002-7147-4604
Rodrigo Fernandez-Gonzalez https://orcid.org/0000-0003-0770-744X
Ashley EE Bruce https://orcid.org/0000-0002-0567-2928

### Ethics

Animal experimentation: This study was performed in strict accordance with the recommendations in the Canadian Council on Animal Care Guidelines on the Care and Use of Fish in Research. All the

animals were handled and maintained according to a protocol (Protocol Number: 20012462) approved by the Biological Sciences Local Animal Care Committee at the University of Toronto.

## Decision letter and Author response
Decision letter https://doi.org/10.7554/eLife.66060.sa1
Author response https://doi.org/10.7554/eLife.66060.sa2

# Additional files
## Supplementary files
• Transparent reporting form

## Data availability
All data generated or analysed during this study are included in the manuscript and supporting files.

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

# Appendix 1

**Appendix 1—key resources table**

| Reagent type (species) or resource | Designation | Source or reference | Identifiers | Additional information |
|---|---|---|---|---|
| chemical compound | rhodamine-phalloidin | Invitrogen | Cat #R415 | (1:200) |
| chemical compound | sytox green | Invitrogen | Cat #S7020 | (1:1000) |
| chemical compound | pHrodo Red Dextran | Invitrogen | Cat# P10361 | (1mg/ml) |
| antibody | anti-phospho-myosin-light chain 2 Ser 19 (Rabbit polyclonal) | Cell Signalling | Cat #3671 | IF(1:100) |
| antibody | anti-Cdh1 (Rabbit polyclonal) | AnaSpec | Cat# 55527s | IF (1:1000) |
| antibody | anti-RAB11B (Rabbit polyclonal) | Abcam | Cat#3612 | IF (1:200) |
| antibody | anti-ZO-1 | ThermoFisher Scientific | Cat#33-9100 ZO1-1A12 | IF (1:500) |
| commercial assay or kit | MEGAscript T7-Transcription Kit | Ambion | Cat#AMB1334 | |
| commercial assay or kit | MegaClear Clean Up Kit | Ambion | Cat#AM1908 | |
| commercial assay or kit | mMessage mMachine T3 Transcription Kit | ThermoFisher Scientific | Cat#AM1348 | |
| commercial assay or kit | NucAway Spin Column | Ambion | Cat#AM10070 | |
| strain (*D. rerio*) | AB wildtype | Zebrafish International Resource Centre | RRID:BDSC_5138 | |
| strain (*D. rerio*) | Tg:(XlEef1a1:dclk2 DeltaK-GFP) | *Sepich et al., 2011* | https://zfin.org/ZDB-TGCONSTRCT-090702-3 | |
| strain (*D. rerio*) | Tg:(XlEef1a:eGFP-tubα8l) | *Fei et al., 2019* | https://zfin.org/ZDB-ALT-170215-4 | Line: uot3Tg |
| strain (*D. rerio*) | Tg:(actb2:myl12.1-eGFP) | *Maître et al., 2012* | https://zfin.org/ZDB-ALT-130108-2 | Line: e2212Tg |
| strain (*D. rerio*) | MZrab25a[4] | This study | https://zfin.org/ZDB-ALT-201221-11 | Line: uot10 |
| strain (*D. rerio*) | MZrab25a[2.3] | This study | https://zfin.org/ZDB-ALT-201221-10 | Line: uot9 |
| strain (*D. rerio*) | MZrab25b | This study | https://zfin.org/ZDB-ALT-201221-12 | Line: uot11 |
| strain (*D. rerio*) | Tg(*actb1:rab25b, myl7:RFP*) | This study | https://zfin.org/ZDB-ALT-220311-4 | Line: uot16Tg |
| strain (*D. rerio*) | MZrab25a[4] Tg:(actb1:myl12.1-eGFP) | This study | | |
| strain (*D. rerio*) | MZrab25b Tg:(actb1:myl12.1-eGFP) | This study | | |
| sequence-based reagent | rab25a_F | This study | PCR Primers | Forward: TATTTATTCACCAAGCGGTTG (for genotyping) |

*Continued on next page*

*Appendix 1—key resources table continued*

| Reagent type (species) or resource | Designation | Source or reference | Identifiers | Additional information |
|---|---|---|---|---|
| sequence-based reagent | rab25a_R | This study | PCR Primers | Reverse: GAGTGGTTCTGGGTGTGAGTC (for genotyping) |
| sequence-based reagent | rab25b_F | This study | PCR Primers | Forward: TGTTTGCAGTGGTTCTTATTGGAG (for genotyping) |
| sequence-based reagent | rab25b_R | This study | PCR Primers | Reverse: ATTACGTTCGCTTGCAGAATTT (for genotyping) |
| sequence-based reagent | rab25a_F | This study | PCR Primers | Forward: ATGGGGACAGATTTAGCCTACAAC (for cDNA) |
| sequence-based reagent | rab25a_R | This study | PCR Primers | Reverse: CGAAGCTGCTGCAAAAACTCCTGA (for cDNA) |
| sequence-based reagent | rab25a_F | This study | PCR primers | Forward: GGATCCATGGGGACAGATTTAGCCTACAAC (ligation into pCS2+ via restriction digest via BamH1, Xho1) |
| sequence-based reagent | rab25a_R | This study | PCR primers | Reverse: CTGGAGCGAAGCTGCTGCAAAAACTCCTGA (ligation into pCS2+ via restriction digest via BamH1, Xho1) |
| sequence-based reagent | rab25a_F | This study | PCR Primers | Forward: CTCGAGGGCGCCACCATGGGGACAGATTTAGCCTACAAC (ligation with into pCS2+ *venus* via Xho1 restriction digest) |
| sequence-based reagent | rab25a_R | This study | PCR Primers | Reverse: CTCGAGCGAAGCTGCTGCAAAAACTCCTGA (ligation with into pCS2+ *venus* via Xho1 restriction digest) |
| sequence-based reagent | rab25b_F | This study | PCR Primers | Forward: GGGGACAAGTTTGTACAAAAAAGCAGGCTTCATGGGGTCTGATGAGGCCTA (*rab25b* with *attb1* for recombination into pDONR221) |
| sequence-based reagent | rab25b_R | This study | PCR Primers | Reverse: GGGGACCACTTTGTACAAGAAAGCTGGGTGTCACAAGTTTTTACAGCAGG (*rab25b* with *attb1* for recombination into pDONR221) |
| sequence-based reagent | rab11a_F | This study | PCR Primers | Forward: AGAAAAACGGTCTGTCCTTC (qPCR) |
| sequence-based reagent | rab11a_R | This study | PCR Primers | Reverse: TCAGGATGGTCTGAAAAGCA (qPCR) |
| sequence-based reagent | rab25a_F | This study | PCR Primers | Forward: GAAGTGACCAGAGGCTCGAT (qPCR) |
| sequence-based reagent | rab25a_R | This study | PCR Primers | Reverse: GGAGTTTTTGCAGCAGCTT (qPCR) |
| sequence-based reagent | rab25b_F | This study | PCR Primers | Forward: TCGGAGCTCTGCTGGTTTAT (qPCR) |
| sequence-based reagent | rab25b_R | This study | PCR Primers | Reverse: GCGTGATCGTAGAGCTCCTT (qPCR) |
| sequence-based reagent | SpvB-F | This study | PCR primers | Forward: ACGATCGATGCCACCATGGGAGGTAATTCATCTCG |
| sequence-based reagent | SpvB-R | This study | PCR primers | Reverse: GACAGGCCTTCATGAGTTGAGTACCCTCA |
| sequence-based reagent | GS1-F | This study | PCR primers | ACGGGATCCGCCACCATGGTGGTGGAACACCCCGA |
| sequence-based reagent | GS1-R | This study | PCR primers | CCGAGGCCTTCAGAATCCTGATGCCACAC |
| sequence-based reagent | Gibson-F | This study | PCR primers | GGTCACTCACGCAACAATACAAGCTACTTGTTCTTTTTG (for cloning from pCS2+ into pFP2) |

*Continued on next page*

*Appendix 1—key resources table continued*

| Reagent type (species) or resource | Designation | Source or reference | Identifiers | Additional information |
|---|---|---|---|---|
| sequence-based reagent | Gibson-R | This study | PCR primers | CATGTCTGGATCATCATTACGTAATACGACTCACTATAG (for cloning from pCS2+ into pFP2) |
| sequence-based reagent | Lsm12_F | This study | PCR Primers | Forward: AGTTGTCCCAAGCCTATGCAATCAG (qPCR) |
| sequence-based reagent | Lsm12_R | This study | PCR Primers | Reverse: CCACTCAGGAGGATAAAGACGAGTC (qPCR) |
| recombinant DNA reagent | pCS2+ (plasmid) | *Rupp et al., 1994* | | SP6/T7 based backbone |
| recombinant DNA reagent | pCS2+egfp-rab25b (plasmid) | This study | | egfp-Rab25b version of pCS2+ |
| recombinant DNA reagent | pCS2+mcherry-rab25b (plasmid) | This study | | mcherry-rab25b version of pCS2+ |
| recombinant DNA reagent | pCS2+venus-rab25a (plasmid) | This study | | venus-rab25a version of pCS2+ |
| recombinant DNA reagent | pCS2+-mcherry-Rab11a (plasmid) | *Rathbun et al., 2020* | | mcherry-Rab11a version of pCS2+ |
| recombinant DNA reagent | pCS2+-mcherry-Mklp1 (plasmid) | *Rathbun et al., 2020* | | mCherry-Mklp1 version of pCS2+ |
| recombinant DNA reagent | pCS2+ lyn-eGfp | A gift from Brian Ciruna | | lyn-eGfp version of pCS2+ |
| recombinant DNA reagent | pTol2-actb1 | A gift from Brian Ciruna | | Tol2 transgenics |
| recombinant DNA reagent | pTol2-actb1:rab25b | This study | | rab25b version of pTol2-actb1 |
| recombinant DNA reagent | pCS2+- Gfp-Utrophin | Addgene | RRID:Addgene_26737 | Gfp-Utrophin version of pCS2+ |
| recombinant DNA reagent | pFP2 | *Narayanan and Lekven, 2012* | | *Wnt8a* enhancer based backbone |
| recombinant DNA reagent | pFP2-GS1 | This study | | GS1 version of FP2 |
| recombinant DNA reagent | pFP2-SPVB | This study | | SPVB version of FP2 |
| recombinant DNA reagent | pFP2-DN-RhoA | This study | | DN-RhoA version of FP2 |
| recombinant DNA reagent | pT3TS-nCas9 | *Jao et al., 2013* | RRID:Addgene_46757 | T3 based backbone |
| recombinant DNA reagent | pSK-H2B-Rfp:5xUAS:Gfp-Dcx | *Distel et al., 2010* | | |

