## [Decision Letter]

**Acceptance summary:**

The authors examined the role of Rab25 during cell division within a developing epithelium. Strikingly, they found that the RabGTPase, Rab25, localized to mitotic structures such as centrosomes and cytokinetic midbodies in dividing cells of the developing zebrafish embryo. They went on to create maternal-zygotic Rab25a and Rab25b mutant embryos where they clearly demonstrate that apical cytokinetic bridges fail to undergo abscission leading to anisotropic cell morphologies that likely contribute to a delayed epiboly. The identification that a Rab11 paralog, Rab25, regulates cytokinesis and abscission when Rab11 is present suggests that Rab25 likely has developed a unique function to Rab11 during abscission.

**Decision letter after peer review:**

Thank you for submitting your article "The recycling endosome protein Rab25 coordinates collective cell movements in the zebrafish surface epithelium" for consideration by *eLife*. Your article has been reviewed by 3 peer reviewers, including Michel Bagnat as the Reviewing Editor and Reviewer #1, and the evaluation has been overseen by Didier Stainier as the Senior Editor.

The reviewers have discussed their reviews with one another, and the Reviewing Editor has drafted this to help you prepare a revised submission. et al.

Essential revisions:

1. Because mz mutants need to be used to examine early embryonic processes it is possible that some of the defects observed are due to impaired processes during oogenesis that may depend or Rab25a or b function.

1A. Are mcherry-Rab25b, Venus-Rab25a and eGFP-Rab25b functional?

1B. Does injection of mRNA for the tagged versions, if functional, or un-tagged Rab25a/b rescue the mz mutant?

2. Genetic compensation is an interesting idea to explain the subtle phenotype. Can mz-rab25a; mz-rab25b double mutants be generated? or at least knock-down rab25b in MZrab25a?

3. Introduce some colocalization studies between Rab25a and/or Rab25b with Rab11 demonstrating that they are on the same compartment during division. Specifically, to compare the spatial and temporal dynamics of both Rab25a and Rab25b with Rab11 and examine the localization of Rab11 in rab25 mutants.

4. Given the seeming strong early phenotype, e.g. multinucleated and highly dysmorphic cells, it is surprising that mz-rab25b survive gastrulation. How consistent is the pattern seen in for example Figure 2C and 3A? do embryos that show that early phenotype survive ? or do surviving embryos present milder early phenotype?

5. The authors suggest Rab25 is implicated in membrane recycling. However, the internalization assay only reflects endocytic events and not recycling.

6A. Could the authors examine the trafficking of membrane proteins involved in cytokinesis?

6B. E-cadherin turn-over is required for cell morphology and cell migration in early zebrafish development. It seems to be worth investigating if the i.e., endocytosis phenotype they observe in MZRab25a/b (Figure 7) is linked to E-cadherin function as the E-cad/half baked mutant phenotype seems to be very similar concerning the EVL (Kane et al.2005).

6C. Could they perform a recycling assay?

[Editors' note: further revisions were suggested prior to acceptance, as described below.]

Thank you for resubmitting your work entitled "The recycling endosome protein Rab25 coordinates collective cell movements in the zebrafish surface epithelium" for further consideration by *eLife*. Your revised article has been evaluated by Didier Stainier as the Senior Editor, and a Reviewing Editor.

Summary:

The authors examined the role of Rab25 during cell division within a developing epithelium. Strikingly, they found that the RabGTPase, Rab25, localized to mitotic structures such as centrosomes and cytokinetic midbodies in dividing cells of the developing zebrafish embryo. They went on to create maternal-zygotic Rab25a and Rab25b mutant embryos where they clearly demonstrate that apical cytokinetic bridges fail to undergo abscission leading to anisotropic cell morphologies that likely contribute to a delayed epiboly. The identification that a Rab11 paralog, Rab25, regulates cytokinesis and abscission when Rab11 is present suggests that Rab25 likely has developed a unique function to Rab11 during abscission.

The reviewers agree in that the manuscript has been significantly improved but there are remaining issues that need to be addressed, as outlined below:

1. The data presented in Figure 7 reporting the intracellular accumulation of Rab11 in rab25 mutants needs quantification to determine whether Rab11+ structures are increased, enlarged or altered in any way in the mutants.

2. Does E-Cadherin accumulate in Rab11 vesicles?

3. It is difficult to appreciate the altered pattern of E-Cadherin the authors describe for Supp. Figure 7. Please provide a clear and quantitative assessment of that phenotype or else remove the data from the manuscript as they currently do not seem to support the claims about it.

If the evidence available is not strong enough to demonstrate membrane recycling defects, claims about such a process being affected should be tempered accordingly as their relative weakness detracts attention from other findings that are well supported and much more compelling.

---

## [Author Response]

Essential revisions:1. Because mz mutants need to be used to examine early embryonic processes it is possible that some of the defects observed are due to impaired processes during oogenesis that may depend or Rab25a or b function.

Given the strong maternal phenotype, as the reviewer notes, we investigated this interesting possibility. We observed mild yolk cell actin defects shortly after fertilization in a small number (N=1/12) of MZ*rab25b* embryos, while all MZ*rab25a* embryos were normal (10/10) at the 4-cell stage. This data is now included in Figure 2—figure supplement 2A and referred to in the text. We observed somewhat abnormal blastoderm morphology in both MZ*rab25a* (36/100) and MZ*rab25b* (30/100) embryos at 3hpf (Figure 2—figure supplement 1D), but these defects resolved by 4.0hpf in the majority of mutant embryos (numbers are now included in the text section titled “Rab25a and Rab25b are required for normal epiboly movements”). These early defects do not appear to explain the EVL defects observed during epiboly based on the findings that (1) EVL morphology is normal at the onset of epiboly in mutants (Figure 2—figure supplement 2C) (2) experimental disruption of yolk actin does not produce the EVL defects seen in mutants (Figure 2—figure supplement 2B and B’). The data can be found in the section titled “Cell Shape and rearrangements indicative of epithelial defects in MZ*rab25a* and MZ*rab25b* embryos.

1A. Are mcherry-Rab25b, Venus-Rab25a and eGFP-Rab25b functional?1B. Does injection of mRNA for the tagged versions, if functional, or un-tagged Rab25a/b rescue the mz mutant?

We apologize if these details were not clear in the text. We had difficulties rescuing the mutant phenotypes by RNA injection, which is likely due to the required maternal contributions of *rab25a* and *rab25b*, as has been observed in other zebrafish maternal mutants (Chen et al., 2017; Li-villarreal et al., 2016; Schwayer et al., 2019). For these reasons we rescued MZ*rab25b* phenotypes using a transgenic rescue construct expressed under the control of a ß-actin promoter (Figure 2—figure supplement 1C). This promoter was previously shown to have high maternal activity making it an ideal approach to rescue maternal mutants (Chen et al., 2017). Therefore, to address the functionality of the tagged Rab25a and Rab25b constructs, maternal mutant lines transgenic for these constructs would have to be generated, which would take in the range of 6-8 months and thus is beyond the scope of this manuscript. Given the documented widespread functionality of N-terminally tagged Rab proteins in zebrafish, mammalian and invertebrate systems (Mavor et al., 2016; Rathbun et al., 2020; Song et al., 2013), we would expect the zebrafish Rab25 constructs to be functional. For example, N-terminally tagged Rab11 has been shown to be functional in the EVL of zebrafish embryos during epiboly (Song et al., 2013) and more recently in Kupfer vesicle formation (Rathbun et al., 2020).

2. Genetic compensation is an interesting idea to explain the subtle phenotype. Can mz-rab25a; mz-rab25b double mutants be generated? or at least knock-down rab25b in MZrab25a?

We now include data in Figure 2—figure supplement 1B showing MZ*rab25a*/*rab25b* double mutant embryos. Double homozygous mutant fish are difficult to generate but the ones we have analyzed consistently display phenotypes resembling MZ*rab25b* embryos with some increase in the severity of the EVL morphology defects. With regards to the issue of genetic compensation, the double mutant phenotype is consistent with our conclusion that Rab25b partially rescues MZ*rab25a* embryos, resulting in a milder phenotype than that seen in MZ*rab25b* embryos. This has now been added to the text.

3. Introduce some colocalization studies between Rab25a and/or Rab25b with Rab11 demonstrating that they are on the same compartment during division. Specifically, to compare the spatial and temporal dynamics of both Rab25a and Rab25b with Rab11 and examine the localization of Rab11 in rab25 mutants.

Given the similarities between Rab25 and Rab11 we agree colocalization experiments would be informative to characterize Rab25’s function. We previously included stills of confocal time lapses of fluorescently taggedRab25a and Rab25b with mCherry-Rab11a in the EVL (Figure 7A, Figure 7—figure supplement 1A, black arrows).

These results showed that Rab25 overlapped with Rab11 positive compartments during epiboly and cell division. To address the temporal comparison, we have added new videos of the time lapses of venus-Rab25a and mCherry-Rab11a during EVL cell division (Video 11). The spatiotemporal association of Rab25 with Rab11 endosomes is similar to observations in mammalian cell culture (Casanova et al., 1999).

4. Given the seeming strong early phenotype, e.g. multinucleated and highly dysmorphic cells, it is surprising that mz-rab25b survive gastrulation. How consistent is the pattern seen in for example Figure 2C and 3A? do embryos that show that early phenotype survive ? or do surviving embryos present milder early phenotype?

We agree that it is surprising that almost all mutant embryos survive. Our live confocal time lapses and phalloidin staining showed that the EVL exhibited highly abnormal cellular phenotypes in nearly 100% of the MZ*rab25b* mutant embryos examined. We have added the data that 99/100 mutant embryos survived gastrulation and thus it not the case that only embryos with mild phenotypes successfully complete gastrulation. How the embryos recover from these severe early defects will be a focus of future studies.

5. The authors suggest Rab25 is implicated in membrane recycling. However, the internalization assay only reflects endocytic events and not recycling.

We have now included results of E-cadherin antibody staining in MZ*rab25b* embryos (Figure 7—figure supplement 1B and C). E-cadherin is a transmembrane protein known to be recycled in the zebrafish EVL (Song et al., 2013), and that therefore can be used to provide some indication of recycling defects. We observed fragmented and reduced E-cadherin at cell-cell contacts in MZ*rab25b* embryos, consistent with a potential recycling defect in mutant embryos. In support of this, we showed that accumulation of lyn-Egfp-positive cytoplasmic membrane compartments in mutant embryos was associated with reduced lyn-egfp fluorescence intensity at the plasma membrane, which could be indicative of reduced fusion of trapped vesicles with the plasma membrane as a consequence of recycling defects. Furthermore, large intracellular Rab11b aggregates shown in Figure 7D are consistent with defective recycling endosome trafficking pathways. These data, taken together with Rab25’s spatiotemporal overlap with Rab11 and the reported role of Rab25 in recycling in other systems, provide some evidence supporting a recycling defect. While we have not performed recycling assays, some of which are technically challenging to do in the zebrafish embryo, we have provided some support for our hypothesis. We make it clear in the text, that our data so far are suggestive but not conclusive of recycling defects in the mutants. We plan to perform FRAP analysis of plasma membrane proteins, including E-cadherin, to more directly address recycling in future studies, which have been delayed due to the pandemic.

6A. Could the authors examine the trafficking of membrane proteins involved in cytokinesis?

Currently, the molecular mechanisms of cytokinesis in zebrafish are not well understood. Thus, while we agree that identifying targets of Rab25 trafficking in cytokinesis is important, we suggest that it is beyond the scope of this manuscript and is something we plan to examine in future work.

6B. E-cadherin turn-over is required for cell morphology and cell migration in early zebrafish development. It seems to be worth investigating if the i.e., endocytosis phenotype they observe in MZRab25a/b (Figure 7) is linked to E-cadherin function as the E-cad/half baked mutant phenotype seems to be very similar concerning the EVL (Kane et al.2005).

See response to comment #5.

6C. Could they perform a recycling assay?

See response to comment #5.

[Editors' note: further revisions were suggested prior to acceptance, as described below.]

[…] The reviewers agree in that the manuscript has been significantly improved but there are remaining issues that need to be addressed, as outlined below:1. The data presented in Figure 7 reporting the intracellular accumulation of Rab11 in rab25 mutants needs quantification to determine whether Rab11+ structures are increased, enlarged or altered in any way in the mutants.

We have now included this data in the section titled “MZ*rab25a* and MZ*rab25b* mutants exhibit endocytic trafficking defects”. We observed increased frequency of Rab11 positive endosomes in EVL cells of MZ*rab25b* embryos (40% of cells, n=46/114, N=8) compared to WT (6% of cells, n=12/206, N=6). This is consistent with the excess of vesicles in MZ*rab25b* embryos observed in Figure 7B, F. The size of Rab11 positive endosomes was also significantly increased in MZ*rab25b* embryos, as shown in the quantification in figure 7—figure supplement 2. We previously observed fluorescently taggedRab25 constructs co-localizing with mCherry-Rab11a in the EVL (Figure 7A). Thus, defects in recycling endosome number and size is consistent with the observed subcellular localization of tagged-Rab25 in the EVL and Rab25’s association with REs in mammalian systems (Casanova et al., 1999; Wang et al., 2000).

2. Does E-Cadherin accumulate in Rab11 vesicles?3. It is difficult to appreciate the altered pattern of E-Cadherin the authors describe for Supp. Figure 7. Please provide a clear and quantitative assessment of that phenotype or else remove the data from the manuscript as they currently do not seem to support the claims about it.

We agree with the reviewers that despite showing increased recycling endosome number and size, our experiments do not directly test whether there are recycling defects in the mutant embryos, which we will explore in future work. Accordingly, as suggested, we have removed the E-cadherin data and tempered our assertions about recycling defects throughout the text.